# KOOPMAN OPERATOR LEARNING FOR ACCELERATING QUANTUM OPTIMIZATION AND MACHINE LEARNING

## ABSTRACT

Finding efficient optimization methods plays an important role for quantum optimization and quantum machine learning on near-term quantum computers. While backpropagation on classical computers is computationally efficient, obtaining gradients on quantum computers is not, because the computational complexity scales linearly with the number of parameters and measurements. In this paper, we connect Koopman operator theory, which has been successful in predicting nonlinear dynamics, with natural gradient methods in quantum optimization. We propose a data-driven approach using Koopman operator learning to accelerate quantum optimization and quantum machine learning. We develop two new families of methods: the sliding window dynamic mode decomposition (DMD) and the neural DMD for efficiently updating parameters on quantum computers. We show that our methods can predict gradient dynamics on quantum computers and accelerate the variational quantum eigensolver used in quantum optimization, as well as quantum machine learning. We further implement our Koopman operator learning algorithms on a real IBM quantum computer and demonstrate their practical effectiveness.

## 1 INTRODUCTION

There has been rapid development of quantum technologies and quantum computation in recent years. A number of efforts are put into demonstrating quantum advantages and speedup. Quantum optimization (Moll et al., 2018) and quantum machine learning (QML) (Biamonte et al., 2017), as important applications of quantum technologies, have received increased interest. The Variational Quantum Eigensolver (VQE) (Peruzzo et al., 2014b; Tilly et al., 2022), as a quantum optimization algorithm, has been developed and applied to understanding problems in high energy physics (Klco et al., 2018; Rinaldi et al., 2022), condensed matter physics (Wecker et al., 2015), and quantum chemistry (Peruzzo et al., 2014a). The Variational Quantum Algorithm (VQA) (Cerezo et al., 2021) such as the Quantum Approximate Optimization Algorithm (QAOA) (Farhi et al., 2014; Harrigan et al., 2021) has been applied to the max-cut problem. A recent experiment on 289 qubits has demonstrated a powerful VQA application in classical optimization by benchmarking against a variety of classical algorithms (Ebadi et al., 2022). QML has also been developed for various tasks including supervised learning (Havlíček et al., 2019), unsupervised learning (Kerenidis et al., 2019) and reinforcement learning (Dong et al., 2008). Theoretical advantages of quantum machine learning have been investigated (Huang et al., 2022b; Liu et al., 2021; 2022), and experiments on real quantum computers have demonstrated encouraging progress (Huang et al., 2022a; Rudolph et al., 2022).

In the noisy intermediate-scale quantum (NISQ) era (Preskill, 2018), due to the noisy nature of current quantum computer architectures, hybrid classical-quantum schemes have been proposed for quantum optimization and quantum machine learning and become a prominent approach. The key spirit of the hybrid approach relies on performing optimization and machine learning on parameterized quantum circuits with quantum features while updating the parameters in the circuit is done through classical computers. In classical machine learning, backprogation only requires vector Jacobian calculations which share the same complexity as the forward evaluation. Obtaining gradients under the hybrid scheme is much more challenging. Calculating gradients in quantum computers is challenging for two reasons: (1) gradient calculation typically scales linearly in the number of parameters as $O(n_{\text{params}})$; and (2) the quantum nature of the gradient itself entails sampling over

repeated measurements. Despite various research on quantum optimization and quantum machine learning in simulation, the implementation of gradient-based methods on real quantum computers is computationally inefficient which limits their applications in practice. It is an important open problem in the field to develop scalable and efficient optimization methods for quantum optimization applications and quantum machine learning tasks.

In this work, we propose Koopman operator learning for accelerating quantum optimization and QML. The Koopman operator theory is a powerful framework for understanding and predicting nonlinear dynamics through linear dynamics embedded into a higher dimensional space (Mezic, 1994; Mezić & Banaszuk, 2004; Mezic, 2005; Rowley et al., 2009; Brunton et al., 2021). By viewing parameter optimization on quantum computers as a nonlinear dynamical evolution in the parameter space, we connect gradient dynamics in quantum optimization to the Koopman operator theory. In particular, the quantum natural gradient helps to provide a natural embedding of original parameters through quantum-computer parameterization into a higher-dimensional space, related to linear imaginary-time evolution. We develop new Koopman operator learning algorithms for quantum optimization and QML, including the sliding window dynamic mode decomposition (SW-DMD) and neural-network-based DMD that learns the Koopman embedding via a neural network parameterization. Our approach is data-driven and based on the information of only a few gradient steps such that the cost of prediction does not scale with $n_{\text{params}}$.

Our methods enable efficient learning of gradient dynamics for accelerating quantum optimization and quantum machine learning. Our experiments are both on numerical simulations and a real quantum computer. We first demonstrate our Koopman operator learning algorithms for VQE, an important application in quantum optimization. We test the methods for the natural gradient and Adam (Kingma & Ba, 2014b) optimizers on quantum Ising model simulations and demonstrate their success on a real IBM quantum computer with a quasi-gradient-based optimizer Simultaneous Perturbation Stochastic Approximation (SPSA) (Spall, 1992). Finally, we apply our methods to accelerate QML on the MNIST dataset.

## 2   RELATED WORK

**Koopman operators.**   Koopman operator theory (Koopman, 1931; v. Neumann, 1932) was first proposed by Koopman and von Neumann in early 1930s to understand dynamical systems. Dynamic mode decomposition (DMD) (Schmid, 2010) was developed to learn the Koopman operator under the linear dynamics assumption of the observed data. Later, more advanced methods such as the extended-DMD based on time-delay embedding (Brunton et al., 2017; Arbabi & Mezic, 2017; Kamb et al., 2020; Tu et al., 2014; Brunton et al., 2016), kernel methods (Baddoo et al., 2022) and dictionary learning (Li et al., 2017) were introduced to go beyond the linear dynamics assumption, and achieved better performance. Recently, machine learning methods were integrated into Koopman operator learning where neural networks are used to learn the mapping to a high dimensional space, in which the dynamics becomes linear (Lusch et al., 2018; Li et al., 2019; Azencot et al., 2020; Rice et al., 2020). The machine learning Koopman operator methods were shown to learn nonlinear differential equation dynamics successfully.

In addition to predicting nonlinear dynamics, recently Koopman operator theory was applied to optimize neural network training (Dogra & Redman, 2020; Tano et al., 2020) and pruning (Redman et al., 2021). These works take the perspective of viewing the optimization process of neural networks as a nonlinear dynamical evolution and uses dynamic mode decomposition to predict the parameter updates in the future. In a more empirical study, Sinha et al. (2017) trained a convolutional neural network (CNN) to predict the future weights of neural networks, trained on standard vision tasks.

Besides important applications for classical systems, the Koopman operator theory has natural connections to quantum mechanics. Recently, researchers considered Koopman operator theory for quantum control (Goldschmidt et al., 2021) and prediction of one particle quantum system evolution (Klus et al., 2022). Since quantum mechanical systems provide a natural high dimensional Hilbert space through the wave function, the theory was considered for embedding classical equations for learning and solving differential equations (Lin et al., 2022; Giannakis et al., 2022).

**Optimization methods on quantum computers.** Due to the nature of hybrid classical-quantum algorithms, taking gradients on quantum computers cannot be done as efficiently as backpropagation on classical computers. In general, the complexity of obtaining gradients scales with the number of parameters on the quantum computer and the number of measurements per parameter. To perform optimization on quantum computers, gradient-free methods such as SPSA and COBYLA (Powell, 1994) are used though they may not scale well to quantum circuits with large number of parameters. For gradient methods, Adam is a common choice for better scaling though its complexity scales with the number of parameters as discussed above. There are also higher order methods such as the quantum natural gradient method which have been shown to have faster convergence while being more challenging to realize experimentally.

**Our work.** We contribute to the development of an efficient data-driven approach that addresses the problems above via a Koopman operator learning approach for accelerating optimization on quantum computers. We integrate the state-of-the-art machine learning Koopman operator with the insight of quantum dynamics to achieve our goal. There are several important features of our work that distinguish it from the relevant works above. First, our goal is to accelerate quantum optimization and quantum machine learning, which compared to classical neural networks has much higher complexity of taking gradients, such that the gain of acceleration will be much more prominent. We further connect the Koopman operator theory with the quantum natural gradient, which does not exist in the classical setups and provides a well-motivated theoretical foundation for our approach. Second, instead of predicting the gradient dynamics directly, we focus on optimizing the loss function. The previous literature (Dogra & Redman, 2020) on Koopman operators for neural networks optimization applies only the standard DMD with no follow-up optimization after the DMD prediction, which may not be well-situated for complicated nonlinear training dynamics. Instead, we develop and investigate the sliding window DMD (SW-DMD) and several variants of neural DMD, including the multi-layer perceptron DMD (MLP-DMD), MLP-SW-DMD and CNN-DMD with an iterative optimization protocol. Our new approach is robust against long-time prediction error and noise, and it is shown to have much better performance compared to the standard DMD. It provides acceleration for optimization in general, and enables novel avenues of research.

## 3 Connecting Koopman Operator Theory and Quantum Optimization

### 3.1 Koopman Operator Theory

We consider a dynamical system with a collection of state variables $\{x(t) \in \mathbb{R}^n\}$ with a transition function $T$ such that $x(t + 1) = T(x(t))$. The Koopman operator theory developed by Koopman asserts that there exists a linear operator $\mathcal{K}$ and a function $g$ such that

$$\mathcal{K}g(x(t)) = g(T(x(t))) = g(x(t+1)) \tag{1}$$

where $\mathcal{K}$ is the Koopman operator. In general, the Koopman operator can act on an infinite-dimensional space. When $\mathcal{K}$ is constrained to a finite dimensional invariant subspace with $g : \mathbb{R}^n \to \mathbb{R}^m$, the Koopman operator can be presented as a Koopman matrix $K \in \mathbb{R}^{m \times m}$.

We now can search for the function $g$. The standard DMD takes $g$ to be the identity function with the assumption that the underlying dynamics of $x$ is approximately linear, *i.e.*, $T$ is a linear operator. The extended-DMD method utilizes other feature functions such as polynomial and trigonometric functions as the basis functions for $g$. To improve that, machine learning methods for the Koopman operator adopt neural networks as universal approximators for learning $g$ (Lusch et al., 2018).

### 3.2 Variational Quantum Eigensolver (VQE)

Consider a Hamiltonian $\mathcal{H}$ describing interactions in a physical system. For a quantum mechanical system with $N$ spins or qubits, $\mathcal{H}$ is a Hermitian operator acting on the $2^N$-dimensional Hilbert space for wave functions. A wave function $\psi$ is an $l_2$-normalized complex-valued vector that contains all the information of the state of the system. In particular, the energy of the system is given by a loss function $\mathcal{L}(\psi) = \langle \psi, \mathcal{H}\psi \rangle$. VQE encodes the wave function as $\psi_{\boldsymbol{\theta}}$ by a set of parameters $\boldsymbol{\theta} \in \mathbb{R}^{n_{\text{params}}}$ via an ansatz layer on a quantum circuit in the quantum computer, as is shown in

the top left part of Figure 1. $n_{\text{params}}$ is usually chosen to be polynomial in $N$, which scales much slower than the $2^N$-scaling of the dimension of $\psi$ itself. The goal of VQE is to minimize the loss by minimizing for $\boldsymbol{\theta}$ in the following objective

$$\boldsymbol{\theta}^* = \arg\min_{\boldsymbol{\theta}} \mathcal{L}(\boldsymbol{\theta}) = \arg\min_{\boldsymbol{\theta}} \langle \psi_{\boldsymbol{\theta}}, \mathcal{H}\psi_{\boldsymbol{\theta}} \rangle. \tag{2}$$

Since the dimension of $\boldsymbol{\theta}$ is usually smaller than the dimension of $\psi$ itself, the loss $\mathcal{L}(\boldsymbol{\theta}^*)$ may still differ from the minimum of $\mathcal{L}(\psi)$. However, thanks to the nonlinearity in the encoding $\psi_{\boldsymbol{\theta}}$ from the nonlinear ansatz layer, in many cases $\boldsymbol{\theta}^*$ can still be a very good approximation of the true minimum of $\mathcal{L}(\psi)$. Quantum machine learning has a similar setup that targets at minimizing $\mathcal{L}(\boldsymbol{\theta})$, and parameterized quantum circuits in QML are called quantum neural networks.

We may use a gradient-based classical optimizer such as Adam to minimize the loss function, and this requires attaining the gradient $\frac{\partial \mathcal{L}}{\partial \theta_i}$. In classical machine learning, the computational cost of backpropagation is only $O(1)$ per iteration. However, in the quantum case, we have to explicitly evaluate the loss with a perturbation in each direction $i$, for example by using the parameter-shift rule (Mitarai et al., 2018; Schuld et al., 2019) $(\mathcal{L}(\theta_i + \pi/2) - \mathcal{L}(\theta_i - \pi/2))/2$, which leads to an $O(n_{\text{params}})$ computational cost per iteration. Hence, quantum optimization is significantly more expensive than classical optimization. As another note, the classical computational components involved in VQE, even including training neural-network-based algorithms in the following sections, typically are much cheaper than the quantum cost.

### 3.3 Quantum Fisher Information and Quantum Natural Gradient

The quantum natural gradient method (Stokes et al., 2020) is a generalization of the classical natural gradient method in classical machine learning (Amari, 1998), where the probability is generalized to the complex-valued wave function. The natural gradient for parameter $\theta$ update for Eq. 2 is given by a nonlinear differential equation

$$\frac{d}{dt}\boldsymbol{\theta}(t) = -\eta F^{-1}\nabla_{\boldsymbol{\theta}}\mathcal{L}(\boldsymbol{\theta}(t)), \tag{3}$$

where $\eta$ is the scalar learning rate, and $F$ is the quantum Fisher Information matrix given by $F_{ij} = \langle \frac{\partial \psi_{\boldsymbol{\theta}}}{\partial \theta_i}, \frac{\partial \psi_{\boldsymbol{\theta}}}{\partial \theta_j} \rangle - \langle \frac{\partial \psi_{\boldsymbol{\theta}}}{\partial \theta_i}, \psi_{\boldsymbol{\theta}} \rangle \langle \psi_{\boldsymbol{\theta}}, \frac{\partial \psi_{\boldsymbol{\theta}}}{\partial \theta_j} \rangle$. It can be shown that the above nonlinear differential equation for $\boldsymbol{\theta}$ is equivalent to the dynamical equation of $\psi_{\boldsymbol{\theta}}(t)$ as follows

$$\frac{d\psi_{\boldsymbol{\theta}}(t)}{dt} = -\mathbb{P}_{\psi_{\boldsymbol{\theta}}}\mathcal{H}\psi_{\boldsymbol{\theta}}(t), \tag{4}$$

where $\mathbb{P}_{\psi_{\boldsymbol{\theta}}}$ is a projector onto the parameterized quantum circuit's manifold (Hackl et al., 2020).

Notice that $\mathcal{H}$ is a linear operator. When the parametrized quantum circuit is sufficiently expressive such that the projection is within the manifold, Eq. 4 can be approximated by a linear differential equation. By viewing the parameters $\boldsymbol{\theta}(t)$ as the state variable $x(t)$ in the Koopman theory, the quantum circuit naturally generates a (wave) function $\psi_{\boldsymbol{\theta}}$ for $\boldsymbol{\theta}$ that plays the role of $g$ in Eq. 1, whose dynamics is close to linear in the Hilbert space. The dimension of $\psi_{\boldsymbol{\theta}}$ is $2^N$, and we do not construct it explicitly. However, the existence of $\psi_{\boldsymbol{\theta}}$ and its approximate linear dynamics build the theoretical foundation for Koopman operator learning algorithms for $\boldsymbol{\theta}(t)$ in optimization and the application of accelerating the parameter updates.

## 4 Koopman Operator Learning Algorithms

### 4.1 Our framework: data-driven approach for accelerating optimization

The algorithmic scheme of Koopman operator learning for quantum optimization and quantum machine learning is shown in Figure 1. A quantum circuit with parameters $\boldsymbol{\theta}$ is used to perform quantum optimization or QML tasks. The loss function $\mathcal{L}$ can be evaluated stochastically through quantum measurements while an optimizer on a classical computer is used to update the parameters. Due to the probabilistic nature of quantum measurements, the measurement precision outcomes usually scales with the standard quantum limit $O(1/\sqrt{n_{\text{shots}}})$ for $n_{\text{shots}}$ quantum measurement shots. To compute the gradient, *e.g.*, by using the parameter-shift rule (Mitarai et al., 2018;

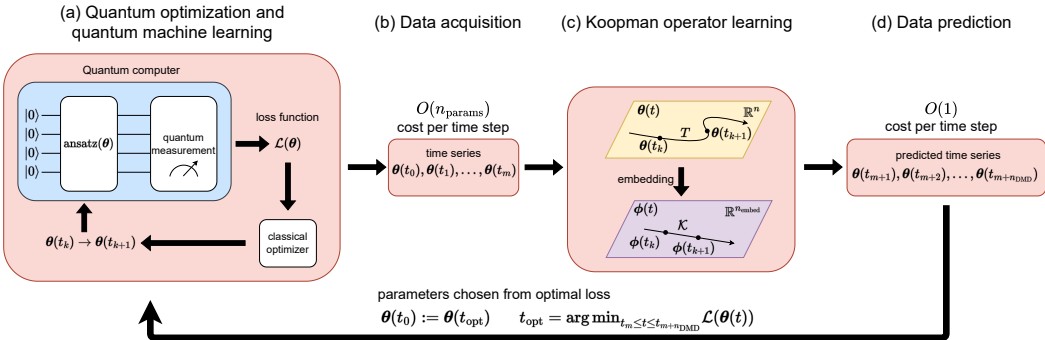

Figure 1: Koopman operator learning for quantum optimization and quantum machine learning. (a) In quantum optimization and QML, a parameterized quantum circuit processes information, and the loss function is evaluated through measurements on a quantum computer. The parameter updates for quantum circuit are computed by a classical optimizer. (b) The optimization history of the parameters forms a time series, where for each time step the gradient optimization complexity scales as $O(n_{\mathrm{params}})$. (c) The Koopman operator learning takes the time series from (b) as training data to find an embedding of the original data with approximately linear dynamics. (d) The Koopman operator predicts the parameter updates where each step has $O(1)$ complexity. The loss from the predicted parameters can be evaluated on quantum computers, and the parameter $\boldsymbol{\theta}(t_{\mathrm{opt}})$ with the optimal loss is used as the starting point for the next iteration in (a).

Schuld et al., 2019), it usually requires quantum measurements' complexity scaling as $O(n_{\mathrm{params}})$ due to the hybrid classical-quantum nature of the algorithm. In addition, the gradient measurements per parameter take $O(n_{\mathrm{shots}})$ complexity, so the total cost of quantum measurement scales as $O(n_{\mathrm{params}} \cdot n_{\mathrm{shots}})$. This is more expensive than the backward propagation in classical machine learning, which has the same complexity as the forward evaluation of the loss function, and the computational cost does not scale as $O(n_{\mathrm{params}})$. After $m$ steps of gradient optimization, a time series of parameter updates $\boldsymbol{\theta}(t_0), \boldsymbol{\theta}(t_1), \ldots, \boldsymbol{\theta}(t_m)$ is obtained.

The Koopman operator learning algorithm takes the time series as a training input to find an embedding in which dynamics becomes approximately linear. It further predicts the parameter updates for $n_{\mathrm{DMD}}$ future steps for the gradient dynamics and obtains $\boldsymbol{\theta}(t_{m+1}), \boldsymbol{\theta}(t_{m+2}), \ldots, \boldsymbol{\theta}(t_{m+n_{\mathrm{DMD}}})$. For parameter predictions in each time step, the parameters can be set in the quantum circuit directly, and the loss function can be evaluated with quantum measurements. This procedure has $O(1)$ cost in terms of the number of parameters, and hence the total cost is $O(n_{\mathrm{shots}})$, which is the same as the forward evaluation of the loss function. Among the $n_{\mathrm{DMD}}$ loss function values $\mathcal{L}(\boldsymbol{\theta}_m)$, we find the lowest loss and the corresponding time $t_{\mathrm{opt}} = \arg\min_{t_m \leq t \leq t_{m+n_{\mathrm{DMD}}}} \mathcal{L}(\boldsymbol{\theta}(t))$. The last VQE iteration $t_m$ is included to avoid degradation, even if the DMD prediction is inaccurate. The optimal Koopman-predicted parameter $\boldsymbol{\theta}(t_{\mathrm{opt}})$ is then used as the initial point for the next quantum optimization or QML step, which completes a full cycle of our algorithm. The alternating VQE+DMD is repeated until the loss reaches the target. Our procedure is robust against long-time prediction error and noise with much fewer gradient measurements, which can efficiently accelerate the gradient-based methods.

## 4.2 SLIDING WINDOW DMD (SW-DMD)

DMD (Brunton et al., 2021) uses a linear fit for the dynamics in the original space for the column vector $\boldsymbol{\theta} \in \mathbb{R}^n$ as $\boldsymbol{\theta}(t_{k+1}) = K\boldsymbol{\theta}(t_k)$. Concatenating $\boldsymbol{\theta}$ at successive times we get data matrices

$$\boldsymbol{\Theta}(t_0) = [\boldsymbol{\theta}(t_0) \quad \boldsymbol{\theta}(t_1) \quad \cdots \quad \boldsymbol{\theta}(t_m)], \quad \boldsymbol{\Theta}(t_1) = [\boldsymbol{\theta}(t_1) \quad \boldsymbol{\theta}(t_2) \quad \cdots \quad \boldsymbol{\theta}(t_{m+1})], \quad (5)$$

where $\boldsymbol{\Theta}(t_1)$ is the one-step time evolution of $\boldsymbol{\Theta}(t_0)$. In the case of approximate linear dynamics, the matrix $K$ is the same for all times $t_k$, and then $\boldsymbol{\theta}(t_{k+1}) = K\boldsymbol{\theta}(t_k)$ extends to $\boldsymbol{\Theta}(t_1) \approx K\boldsymbol{\Theta}(t_0)$, where $K \in \mathbb{R}^{n \times n}$. The best fit is at the minimum of the Frobenius loss ($^+$ is the pseudo-inverse):

$$K = \arg\min_K \|\boldsymbol{\Theta}(t_1) - K\boldsymbol{\Theta}(t_0)\|_F = \boldsymbol{\Theta}(t_1)\boldsymbol{\Theta}(t_0)^+. \quad (6)$$

Figure 2: Neural network architectures for our neural DMD approaches. (a) MLP bottleneck architecture with MSE loss for training. (b) CNN bottleneck architecture that operates on simulations as temporal dimension and parameters as channel dimension.

When the dynamics of $\boldsymbol{\theta}$ is not linear, we can instead consider a time-delay embedding with a sliding window and concatenate the steps to form an extended data matrix (Dylewsky et al., 2022)

$$\boldsymbol{\Phi}(\boldsymbol{\Theta}(t_0)) = [\boldsymbol{\phi}(t_0) \quad \boldsymbol{\phi}(t_1) \quad \cdots \quad \boldsymbol{\phi}(t_m)] = \begin{bmatrix} \boldsymbol{\theta}(t_0) & \boldsymbol{\theta}(t_1) & \cdots & \boldsymbol{\theta}(t_m) \\ \boldsymbol{\theta}(t_1) & \boldsymbol{\theta}(t_2) & \cdots & \boldsymbol{\theta}(t_{m+1}) \\ \vdots & \vdots & \ddots & \vdots \\ \boldsymbol{\theta}(t_d) & \boldsymbol{\theta}(t_{d+1}) & \cdots & \boldsymbol{\theta}(t_{m+d}) \end{bmatrix}. \tag{7}$$

$\boldsymbol{\Phi}$ is generated by a sliding window of size $d+1$ at $m+1$ consecutive time steps. Each column of $\boldsymbol{\Phi}$ is a time-delay embedding for $\boldsymbol{\Theta}$, and the different columns $\boldsymbol{\phi}$ in $\boldsymbol{\Phi}$ are embeddings at different starting times. The time-delay embedding captures some nonlinearity in the dynamics of $\boldsymbol{\theta}$, with $\boldsymbol{\Theta}(t_{d+1}) \approx K\boldsymbol{\Phi}(\boldsymbol{\Theta}(t_0))$, where $K \in \mathbb{R}^{n \times n(d+1)}$. The best fit is given by

$$K = \arg\min_K \|\boldsymbol{\Theta}(t_{d+1}) - K\boldsymbol{\Phi}(\boldsymbol{\Theta}(t_0))\|_F = \boldsymbol{\Theta}(t_{d+1})\boldsymbol{\Phi}(\boldsymbol{\Theta}(t_0))^+. \tag{8}$$

The used data from the acquired time series with the largest time in the above equation is $\boldsymbol{\theta}(t_{m+d+1})$. During prediction we start with $\boldsymbol{\theta}(t_{m+d+2}) = K\boldsymbol{\phi}(t_{m+1})$. Then we update from $\boldsymbol{\phi}(t_{m+1})$ to $\boldsymbol{\phi}(t_{m+2})$ by removing the oldest data $\boldsymbol{\theta}(t_{m+1})$ and adding the newly predicted data $\boldsymbol{\theta}(t_{m+d+2})$. We iteratively repeat prediction via $\boldsymbol{\theta}(t_{k+d+1}) = K\boldsymbol{\phi}(t_k)$. Our approach is different from that of Dylewsky et al. (2022), as we do not use an additional SVD before doing DMD and our matrix $K$ is non-square. We denote DMD performed this way as sliding window DMD (SW-DMD). The standard DMD is a special case of SW-DMD when the sliding window size is 1 (*i.e.*, $d = 0$).

### 4.3 NEURAL DMD

**General formulation.** To provide a better approximation to the nonlinear dynamics, we ask whether the hard-coded sliding window transformation $\boldsymbol{\Phi}$ can be a neural network. Thus, by simply reformulating $\boldsymbol{\Phi}$ in Eq. 8 as a neural network, we formulate a natural neural objective for Koopman operator learning as follows

$$\arg\min_{K,\alpha} \|\boldsymbol{\Theta}(t_{d+1}) - K\boldsymbol{\Phi}_\alpha(\boldsymbol{\Theta}(t_0))\|_F, \tag{9}$$

where $K \in \mathbb{R}^{N_{in} \times N_{out}}$ is a linear Koopman operator and $\boldsymbol{\Phi}_\alpha(\boldsymbol{\Theta}(t_0))$ is a nonlinear neural embedding by a neural network $\boldsymbol{\Phi}_\alpha$ with parameters $\alpha$. $\boldsymbol{\Phi}_\alpha := \mathrm{NN}_\alpha \circ \boldsymbol{\Phi}$ is a composition of the neural network architecture $\mathrm{NN}_\alpha$ and the sliding window embedding $\boldsymbol{\Phi}$ from the previous section.

**MLP-DMD, CNN-DMD and MLP-SW-DMD methods** Inspired by the machine learning advancement of DMD (Lusch et al., 2018), we introduce *MLP-DMD*, which uses a simple MLP architecture for $\boldsymbol{\Phi}$, as shown in Figure 2 (a). The architecture consists of two linear layers with an ELU (Clevert et al., 2015) activation and a residual connection. We explored an expansion ratio within $\{1,2\}$ for the hidden layer, but selected 1 as the best one, possibly because higher ratios present a risk of overfitting to the noise in the history of the optimization. However, it does not use mixing of information between simulation steps because the simulation steps are on the batch dimension. We can also consider an 1D CNN encoder in Figure 2 (b) to form the *CNN-DMD* method. In CNN-DMD the simulation steps form the temporal dimension and the parameters form

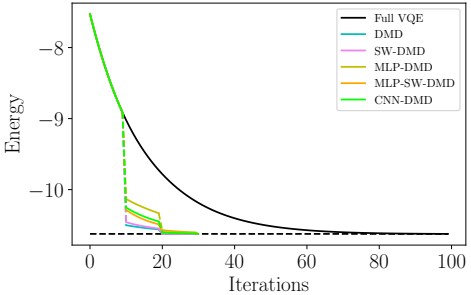
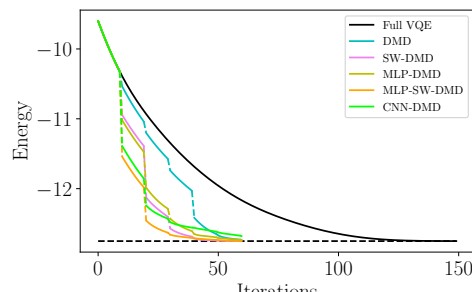

(a) Natural gradient 10-qubit results with $n_{\text{sim}} = 10$, $n_{\text{DMD}} = 40$, and $n_{\text{SW}} = 6$ for SW-DMD and MLP-SW-DMD. For the various DMD methods, the solid piecewise curves are actual gradient steps, and the dashed lines connecting them indicate when the DMD prediction is applied.

(b) Adam 12-qubit results with $n_{\text{sim}} = 10$, $n_{\text{DMD}} = 40$, $n_{\text{SW}} = 6$ for SW-DMD and MLP-SW-DMD. For the various DMD methods, the solid piecewise curves are actual gradient steps, and the dashed lines connecting them indicate when the DMD prediction is applied.

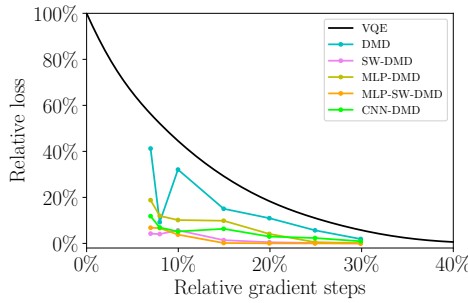
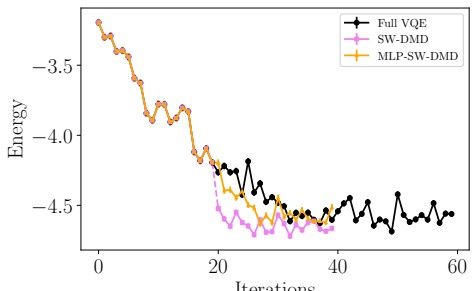

(c) Adam 12-qubit results for relative loss versus relative gradient steps. Less relative gradient steps mean less quantum resource used. Lower relative loss indicates better performance of the optimization. See Table. 1 for more details.

(d) Experimental results for 5-qubit Ising model using SPSA from the real quantum computer IBM Lima. Statistical errorbars are plotted but are relatively small with 10,000 shots.

Figure 3: Experimental results for the quantum Ising model at $h = 0.5$.

the channel dimension of the CNN. We use causal masking of the CNN kernels on the encoder to utilize parameters information in all previous steps without look-ahead bias. During the inference we look at the history of simulated steps, and look at the prediction of the last step. Then we recurrently feed the last step and resume the predictions. Our architecture consists of two 1D-CNN layers with bottleneck middle channel number of 1 (to avoid overfitting) and ELU activation in between. See Appendix C for more details. We also define an *MLP-SW-DMD* method which is similar to MLP-DMD but applies the time-delay embedding to the input. MLP-DMD and CNN-DMD both generalize from DMD, and MLP-SW-DMD generalizes from SW-DMD.

## 5 EXPERIMENTS

### 5.1 QUANTUM OPTIMIZATION

For quantum optimization, we perform experiments for the 1D quantum Ising model with the transverse field $h$ defined in Appendix B. The goal is to find the wave function that minimizes the corresponding Hamiltonian $\mathcal{H}$. We use $h = 0.5$ in this section, and show experimental results for $h = 0.9, 2.0$ in Appendix I. We also implement experiments for the quantum Heisenberg model with results in Appendix H. For each Hamiltonian, we perform a full VQE for $n_{\text{total}}$ iterations, and our alternating VQE+DMD algorithms with $n_{\text{sim}}$ VQE iterations and $n_{\text{DMD}}$ DMD prediction steps in each piece. Detailed setups of the alternating VQE+DMD are given in Appendix A.

### 5.1.1 NOISELESS QUANTUM SIMULATIONS

**Noiseless quantum natural gradient simulations.** In Figure 3a, in the case of using natural gradient, we show the acceleration effects of the DMD methods on a 10-qubit Ising model with the circular-entanglement RealAmplitudes ansatz and reps=1 (2 layers, 20 parameters) explained in Appendix D. The learning rate is $0.01$. We perform $n_{\text{total}} = 100$ pure VQE iterations. On the DMD curves, the piecewise solid lines are from actual gradient steps, and the dashed lines connecting them are where the DMD algorithms are used to find $\boldsymbol{\theta}(t_{\text{opt}})$. Details of the predicted loss from DMD and the effect of $n_{\text{SW}}$ in SW-DMD are discussed in Appendices E and G respectively. All the DMD methods are able to significantly accelerate the quantum optimization, so that the energy at 30 iterations is comparable to the final full VQE loss at 100 iterations. DMD and SW-DMD perform better than MLP-DMD, MLP-SW-DMD, and CNN-DMD, possibly because of the potential overfitting of the neural networks since the parameters dynamics with natural gradient is theoretically relatively simple according to Sec. 3.3. Results for the quantum Heisenberg model are in Appendix H.

**Noiseless Adam simulations.** Instead of using the natural gradient method, we show another application by using the Adam (Kingma & Ba, 2014a) optimizer based on the gradients obtained from the parameter-shift rule. In real quantum experiments, it is much less challenging to implement Adam than the quantum natural gradient method. We perform experiments on the 12-qubit Ising model with the circular-entanglement RealAmplitudes ansatz and reps=1 (2 layers, 24 parameters). In Figure 3b, we choose $n_{\text{total}} = 150$, $n_{\text{sim}} = 10$, $n_{\text{DMD}} = 40$, and for SW-DMD and MLP-SW-DMD, $n_{\text{SW}} = 6$. The Adam learning rate is $0.01$. While theoretically the prediction of the Adam update could be more complicated than the natural gradient, the energies with various DMD methods at 60 iterations are comparable to the full VQE final energy at 150 iterations, and the acceleration is very significant. In particular, SW-DMD and MLP-SW-DMD converge quickly.

### 5.1.2 ABLATIONS

To measure the acceleration effect, we implement a series of ablation experiments. We use the full VQE as a reference. For DMD methods, we define *Relative gradient steps* as the number of gradient steps that DMD methods utilize compared to the full VQE gradient steps, and *Relative loss* $= (L_{\text{min, VQE + DMD}} - L_{\text{min, full VQE}})/(L_{\text{initial, full VQE}} - L_{\text{min, full VQE}})$, where $L_{\text{min, VQE + DMD}}$, $L_{\text{min, full VQE}}$ are the minimum loss for VQE+DMD and full VQE respectively, $L_{\text{initial, full VQE}}$ is the initial loss of full VQE. The cost per gradient step has a $O(n_{\text{params}})$-scaling, and is the dominant cost for quantum optimization. Therefore, we use the relative gradient steps as the main metric for the cost of quantum resources. A smaller relative loss means better performance of the optimization.

We consider the same 12-qubit Ising model as the noiseless Adam simulations in Sec. 5.1.1. With $n_{\text{DMD}} = 90$, and $n_{\text{SW}} = 6$ for SW-DMD and MLP-SW-DMD, we show ablations associated with varying $n_{\text{sim}}$. $n_{\text{DMD}}$ is chosen large enough so that this hyperparameter does not affect significantly the minimum loss. In Figure 3c, we show the relative loss versus the relative gradient steps (See Appendix F Table. 1 for details). The DMD methods should be compared to the performance of pure VQE (black line). All the DMD methods are able to accelerate quantum optimization, as in Figure 3c, their relative loss curves are below the full VQE curve. SW-DMD works best for relative gradient steps $< 10\%$, and MLP-SW-DMD works best for relative gradient steps $\geq 10\%$.

SW-DMD adds the sliding-window embedding to DMD, and improves the performance significantly, compared to DMD. MLP-DMD and CNN-DMD both add the neural networks to DMD, and in most cases perform better than DMD. Likewise, MLP-SW-DMD adds the neural networks to SW-DMD, and yields better performance than SW-DMD when the relative gradient steps are relatively high. In lower relative gradient steps regimes, since $n_{\text{sim}}$ is lower, *i.e.*, less time series data for training, adding the neural networks may be less beneficial.

### 5.1.3 QUANTUM OPTIMIZATION ON A REAL QUANTUM COMPUTER

To test our approach on real quantum hardware, we further implement the VQE and DMD methods on the 5-qubit Ising model with $h = 0.5$ on a real IBM quantum computer Lima. The experiments are performed with the 2-layer linear-entanglement RealAmplitudes ansatz (10 parameters), SPSA (learning rate $0.04$ and perturbation $0.1$), 10k quantum shots, and measurement error mitigation.

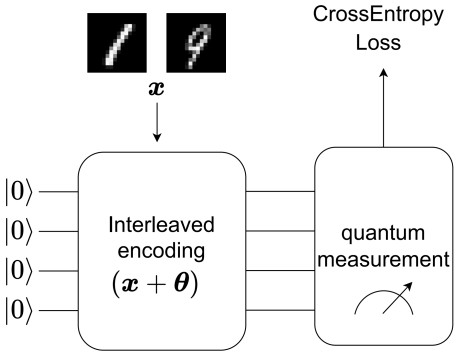

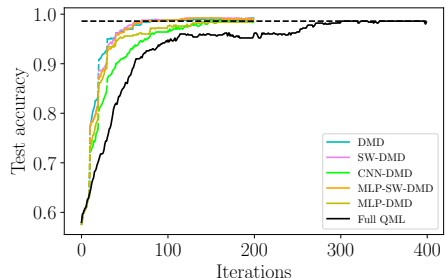

(a) Quantum machine learning architecture with interleaved encoding of 10-qubit quantum circuit for a binary classification task on MNIST.

(b) Accuracy of binary classification for full QML simulation and DMD methods. For the various DMD methods, solid piecewise curves are from actual gradient steps, and the dashed lines connecting them indicate the use of DMD prediction. See Table. 3 for more details.

Figure 4: Quantum machine learning architecture and results.

As is shown in Figure 3d, on IBM Lima, we perform full VQE for 60 iterations. SW-DMD and MLP-SW-DMD use $n_{\mathrm{SW}} = 15$. We use $n_{\mathrm{sim}} = 20$, $n_{\mathrm{DMD}} = 20$. We only run the SW-DMD and MLP-SW-DMD methods, because other methods already have unstable performances with the same setups from the noise model simulation on FakeLima which mimics the real Lima, and real quantum computations are costly. With SW-DMD, the optimization is successfully accelerated, but with MLP-SW-DMD, the acceleration is less significant. The algorithmically simpler method, SW-DMD, may have more stability and robustness in a realistic setup. Appendix K discusses more details and analysis, and Appendix J contains more simulations with quantum noise effects.

## 5.2 QUANTUM MACHINE LEARNING

Our approach is also applied to accelerate quantum machine learning. We consider the task of binary classification on a filtered MNIST dataset with samples labeled by digits "1" and "9". We use an interleaved block-encoding scheme for QML, which is shown to have generalization advantage (Jerbi et al., 2021; Caro et al., 2021; Li et al., 2022; Ren et al., 2022b) and recently realized in experiment (Ren et al., 2022a). The quantum neural network is shown in Figure 4a and we apply the Koopman operator learning to accelerate the learning process. Each image is first downsampled to $16 \times 16$ pixels, and then used as an input $\boldsymbol{x} \in [0, 1]^{256}$ that is fed into an interleaved encoding quantum gate,. The parameters $\boldsymbol{\theta}$ are also encoded in the interleaved encoding quantum gate. Then the quantum measurements are used as the output for computing the cross-entropy loss. We simulate a 10-qubit quantum computer with $n_{\mathrm{params}} = 270$. The details of the QML data and architecture are in Appendix L. To reduce the size of inputs of the neural networks in MLP-DMD, MLP-SW-DMD, CNN-DMD, we adopt a layerwise partitioning in $\boldsymbol{\theta}$ with details in Appendix L. Figure 4b shows that all DMD methods can achieve good accuracy at 150 to 200 iterations while saving quantum resources compared to full QML at 300 to 400 iterations, which indicates significant acceleration.

**Conclusion.** Quantum optimization and quantum machine learning have the potential of being powerful tools for finding solutions of certain problems that are difficult for classical algorithms. However, gradient-based optimization on quantum computers takes $O(n_{\mathrm{params}})$ gradient measurements for each iteration, and thus can be very expensive, especially in the current era with limited quantum computing resources. Anchored in a theoretical connection between the Koopman operator theory and the quantum natural gradient, we propose a data-driven approach to use Koopman learning for accelerating quantum optimization and quantum machine learning. We further use the power of the time-delay embedding and neural networks to address the nonlinearity in optimization and achieve better prediction, which brings down the expense of quantum optimization and quantum machine learning both on simulated and real quantum computer experiments. Our work serves as a bridge between the machine learning and quantum optimization communities and opens up new opportunities to explore the efficiency of optimization problems through Koopman operator theory.

## 6 REPRODUCIBILITY OF RESULTS

We have stated all of our hyperparameter choices for the experimental settings in the main text and in the appendix. We perform simulations of VQE using Qiskit (ANIS et al., 2021), a python framework for quantum computation. Our neural network code is based on Qiskit and Pytorch (Paszke et al., 2019). Our implementation of quantum machine learning is based on Yao (Luo et al., 2020), a framework for quantum algorithms in Julia (Bezanson et al., 2017). We will release our code on Github upon acceptance.

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

## A  SETUP OF VQE+DMD RUNS

In quantum optimization, we start from a random initial point in the parameter space and run VQE for $n_{\text{sim}}$ iterations. Then, we apply various DMD methods to the trajectory of parameters from VQE, predict the future trajectory for $n_{\text{DMD}}$ steps, and evaluate the energies on the predicted trajectory. From the $n_{\text{DMD}}$ energy evaluations, the optimal parameters corresponding to the minimum energy are used as the initial point of the next $n_{\text{sim}}$ iterations of VQE. Without particular specification, the word "iteration" in this work usually refers to a step that requires gradient evaluation. Since per gradient cost scales as $O(n_{\text{params}})$, it is dominant over the DMD prediction cost, and the number of iterations indicates the main cost of the algorithm. We repeat the alternating runs of VQE and DMD with $n_{\text{sim}}$ and $n_{\text{DMD}}$ as hyperparameters. In SW-DMD and MLP-SW-DMD, the sliding window size $n_{\text{SW}}$ is an additional hyperparameter with the requirement $n_{\text{SW}} < n_{\text{sim}}$. DMD is a special case of SW-DMD with $n_{\text{SW}} = 1$. We benchmark the standard DMD, SW-DMD, MLP-DMD, MLP-SW-DMD and CNN-DMD with $n_{\text{sim}} = 10$. The sliding window size is chosen as $n_{\text{SW}} = 6$ for SW-DMD and MLP-SW-DMD. (We use $n_{\text{sim}} = 20$ and $n_{\text{SW}} = 15$ on the real quantum computer SPSA experiments shown in Sec. 5.1.3.) We also perform a pure VQE run for $n_{\text{total}}$ iterations, starting from the same initial point as the alternating VQE+DMD runs.

## B  QUANTUM ISING MODEL AND QUANTUM HEISENBERG MODEL

We first introduce the Pauli matrices $X$, $Y$ and $Z$

$$X = \begin{bmatrix} 0 & 1 \\ 1 & 0 \end{bmatrix}, \quad Y = \begin{bmatrix} 0 & -i \\ i & 0 \end{bmatrix}, \quad Z = \begin{bmatrix} 1 & 0 \\ 0 & -1 \end{bmatrix}. \tag{10}$$

The quantum Ising model with transverse field has the following Hamiltonian

$$\mathcal{H} = -\sum_{i=1}^{N} Z_i \otimes Z_{i+1} - h \sum_{i=1}^{N} X_i. \tag{11}$$

The quantum Heisenberg has the following Hamiltonian

$$\mathcal{H} = \sum_{i=1}^{N} \left( X_i \otimes X_{i+1} + Y_i \otimes Y_{i+1} \right) + J_z \sum_{i=1}^{N} Z_i \otimes Z_{i+1}. \tag{12}$$

The subscripts on the Pauli matrices denote which qubit they are acting on. For both models, we use the periodic boundary condition such that the qubits at $i = 1$ and $i = N + 1$ are identical.

## C  FURTHER DETAILS ON NEURAL DMD

**Optimization**  The parameters $K$ and $\alpha$ are trained jointly on Eq. 9 from scratch with every new batch of optimization history by using the Adam optimizer for 30k steps with 9k steps of linear warmup from 0 to 0.001 and then cosine decay back to 0 at step 30k. We use the MSE loss, which minimizes the Frobenius norm.

**Activation**  Along with ELU, we also explored cosine, ReLU and tanh activations. We found ELU to be the best, likely because: tanh suffers from vanishing gradients, ReLU biases to positive numbers (while input phases quantum circuit parameters $\boldsymbol{\theta}$ are unconstrained) and cosine is periodic.

**The importance of learning rate scheduler.**  We train the neural network $\boldsymbol{\Phi}_\alpha$ from scratch every time we obtain optimization history as training data. It is important that we have a stable learning with well converging neural network at every single Koopman operator fitting stage. For that purpose, we found it vital to use a cosine-decay scheduler with a linear warmup, which is typically useful in the computer vision literature (Loshchilov & Hutter, 2016). Namely, for the first 9k steps of the neural network optimization, we linearly scale the learning rate from 0 to 0.001, and then use a cosine-decay from 0.001 to 0 until the final step at 30k. In Sec. 5.2 for quantum machine learning, we use 80k training steps for CNN to achieve a better performance.

**The importance of residual connections.** In our work we use a residual connection in order to make DMD as a special case of the neural DMD parameterization. The residual connection is indeed very useful, as it is driven by the Koopman operator learning formulation. Namely, the residual connection makes it possible for the encoder to learn the identity. If the encoder becomes the identity, then MLP(-SW-)DMD or CNN-DMD become vanilla (SW-)DMD.

**The procedure for making predictions using neural DMD.** Eq. 9 provides the general objective $\arg\min_{K,\alpha} \|\mathbf{\Theta}(t_{d+1}) - K\mathbf{\Phi}_\alpha(\mathbf{\Theta}(t_0))\|_F$ for training the neural-network DMD including MLP-DMD, CNN-DMD, and MLP-SW-DMD. First, for MLP-DMD and CNN-DMD, we take $d = 0$ with no sliding window, so the objective gets reduced to $\arg\min_{K,\alpha} \|\mathbf{\Theta}(t_1) - K\mathbf{\Phi}_\alpha(\mathbf{\Theta}(t_0))\|_F$. In the phase of training, the optimization history $\boldsymbol{\theta}(t_0), \boldsymbol{\theta}(t_1), ..., \boldsymbol{\theta}(t_{m+1})$ is first concatenated into $\mathbf{\Theta}(t_0)$ and $\mathbf{\Theta}(t_1)$ in the same way as in SW-DMD using Eq. 5. In every column, $\mathbf{\Theta}(t_1)$ is one iteration ahead of $\mathbf{\Theta}(t_0)$ in the future direction. Then, in training, the operator $\mathbf{\Phi}_\alpha = \mathrm{NN}_\alpha$, as a neural network architecture, acts only on $\mathbf{\Theta}(t_0)$ not on $\mathbf{\Theta}(t_1)$. $K$ is a square Koopman matrix that denotes a forward dynamical evolution from $\mathbf{\Phi}_\alpha\mathbf{\Theta}(t_0)$ directly to $\mathbf{\Theta}(t_1)$, rather than from $\mathbf{\Phi}_\alpha\mathbf{\Theta}(t_0)$ to $\mathbf{\Phi}_\alpha\mathbf{\Theta}(t_1)$. Likewise, in the phase of inference for predicting the future of $\boldsymbol{\theta}$ beyond $t_{m+1}$, we apply the operator $K\mathbf{\Phi}_\alpha$ repeatedly using the evolution $\mathbf{\Theta}(t_{k+1}) = (K\mathbf{\Phi}_\alpha)^k\mathbf{\Theta}(t_1)$, without an explicit inversion of the operator $\mathbf{\Phi}_\alpha$ to bring back the original representation. Next, for MLP-SW-DMD, we need to put $d$ back to the equations and make the operator $\mathbf{\Phi}_\alpha = \mathrm{NN}_\alpha \circ \mathbf{\Phi}$ a composition of the neural network architecture $\mathrm{NN}_\alpha$ and the sliding window embedding $\mathbf{\Phi}$. The Koopman operator $K$ of MLP-SW-DMD has the same dimension as $K$ of SW-DMD, which is non-square. The procedure of updating $\mathbf{\Phi}(\mathbf{\Theta}(t))$ by adding the latest data and removing the oldest data is the same as in SW-DMD described in Sec. 4.2. The only difference between MLP-SW-DMD and SW-DMD is the additional neural network architecture $\mathrm{NN}_\alpha$ in MLP-SW-DMD. The only difference between MLP-SW-DMD and MLP-DMD is the additional sliding window embedding $\mathbf{\Phi}$ in MLP-SW-DMD.

## D  REALAMPLITUDES ANSATZ

The RealAmplitudes ansatz from Qiskit is an ansatz that always produces a real-valued wave function $\psi_{\boldsymbol{\theta}}$ without the imaginary part. The minimum loss of the Ising-model Hamiltonian is always achievable by a real-valued $\psi_{\boldsymbol{\theta}}$, so this ansatz can reduce the redundancy in the functional form of $\psi_{\boldsymbol{\theta}}$ and is beneficial for our use.

$\psi_{\boldsymbol{\theta}}$ as a function of $\boldsymbol{\theta}$ is a nonlinear function, which consists of alternating layers of a rotational-$Y$ gate $R_Y(\theta)$,

$$R_Y(\theta) = \exp\left(-i\theta Y/2\right) = \begin{bmatrix} \cos\left(\theta/2\right) & -\sin\left(\theta/2\right) \\ \sin\left(\theta/2\right) & \cos\left(\theta/2\right) \end{bmatrix} \tag{13}$$

followed by a 2-qubit controlled-$X$ gates with no parameter

$$CX = \begin{bmatrix} 1 & 0 & 0 & 0 \\ 0 & 0 & 0 & 1 \\ 0 & 0 & 1 & 0 \\ 0 & 1 & 0 & 0 \end{bmatrix}. \tag{14}$$

Each layer of the $R_Y(\theta)$ gates has $N$ (the number of qubits) parameters, so the total number of parameters is equal to the number of rotational layers times $N$.

## E  DMD PREDICTIONS

In this section, we show the details of the DMD predictions that we have used to accelerate VQE on the quantum Ising model at $h = 0.5$ with natural gradient and Adam presented in Figures 3a and 3b respectively.

Figure 5ashows the natural gradient results with DMD predictions. For the various DMD methods, solid parts are VQE runs, and the dashed parts are DMD predictions. The horizontal axis label is the total steps including both the actual gradient steps and prediction steps. For the pure VQE, the total steps are equal to the iterations, which are all actual gradient steps. For the DMD curves, the solid

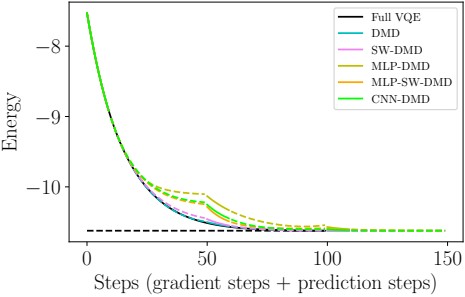

(a) Natural gradient 10-qubit results with $n_{\mathrm{sim}} = 10$, $n_{\mathrm{DMD}} = 40$, and $n_{\mathrm{SW}} = 6$ for SW-DMD and MLP-SW-DMD. For the various DMD methods, solid parts are VQE runs, and the dashed parts are DMD predictions.

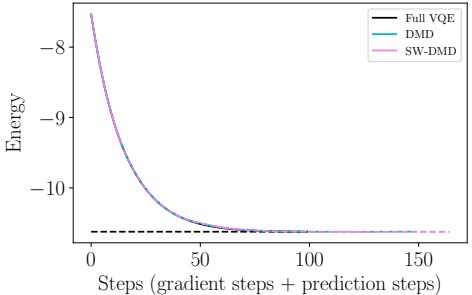

(b) Comparison in natural gradient 10-qubit results for $n_{\mathrm{sim}} = 10$ with DMD and $n_{\mathrm{sim}} = 15$, $n_{\mathrm{SW}} = 6$ with SW-DMD. Both methods use $n_{\mathrm{DMD}} = 40$. For both DMD methods, solid parts are VQE runs, and the dashed parts are DMD predictions.

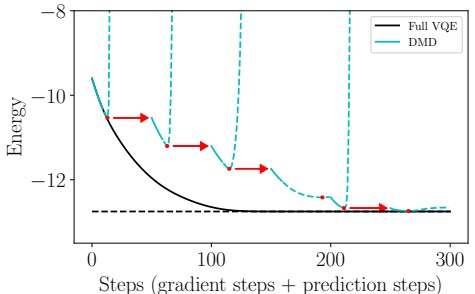

(c) Adam 12-qubit results with the standard DMD method with $n_{\mathrm{sim}} = 10$, $n_{\mathrm{DMD}} = 40$. For the DMD method, solid parts are VQE runs, and the dashed parts are DMD predictions. The red points mark the optimal parameters for each piece of DMD predictions. The next piece of VQE starts from the optimal parameters rather than the last parameters, as is indicated by the red arrows, to guarantee the decrease of energy.

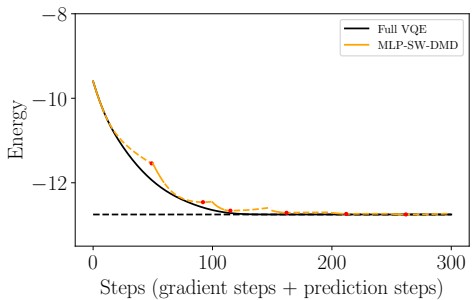

(d) Adam 12-qubit results with the MLP-SW-DMD method with $n_{\mathrm{sim}} = 10$, $n_{\mathrm{DMD}} = 40$. For the DMD method, solid parts are VQE runs, and the dashed parts are DMD predictions. The red points mark the optimal parameters for each piece of DMD predictions, as the initial point for the next piece of VQE.

Figure 5: Experimental results with DMD predictions displayed. The horizontal axis label is the total steps, including the gradient steps and the prediction steps.

parts are gradient steps with $O(n_{\mathrm{params}})$ cost per step, and the dashed parts are prediction steps with $O(1)$ cost per step. Since with the natural gradient, $\boldsymbol{\theta}$ has the quantum wave function as a natural embedding with dynamics close to be linear, Koopman operator learning is hypothesized to yield good predictions. All the DMD algorithms are able to achieve the full VQE optimal loss. DMD and SW-DMD both match the full VQE. Our MLP-DMD, MLP-SW-DMD, and CNN-DMD start to deviate from the full VQE when the DMD iterations increase. This could be related to overfitting of the neural network when the parameters dynamics are simple. With $n_{\mathrm{sim}} = 10$ both for DMD and SW-DMD ($n_{\mathrm{SW}} = 6$), DMD has a slight better performance when make predictions, this can be because in SW-DMD, there are only $n_{\mathrm{sim}} - n_{\mathrm{SW}} + 1 = 5$ columns of data in the time-delay embedding, fewer than 10 independent columns of the standard DMD (equivalent to $n_{\mathrm{SW}} = 1$). However, when we use $n_{\mathrm{sim}} = 15$ and $n_{\mathrm{SW}} = 6$ for SW-DMD resulting in 10 columns in the time-delay embedding, the performance of SW-DMD is comparable to the standard DMD, as is shown in Figure 5b.

In Figure 5c, we show the standard DMD predictions in the Adam case. The dashed curves are from DMD predictions, which typically first lower the energy but then lead to energy explosion. The energy explosion, however, does not make our algorithm fail, because we start the next piece of

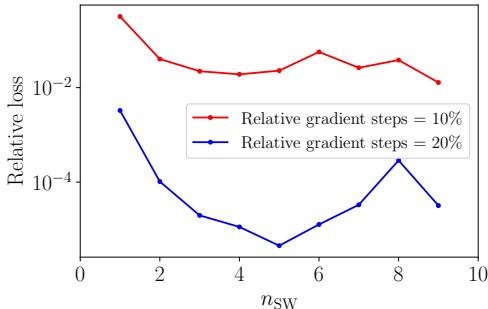

Figure 6: Relative loss from SW-DMD versus $n_{\text{SW}}$ at $10\%$ and $20\%$ relative gradient steps in the range $1 \leq n_{\text{SW}} \leq 9$. The expriments are all performed on the 12-qubit quantum Ising model at $h = 0.5$ . We use $n_{\text{total}} = 300$ full VQE iterations, $n_{\text{sim}} = 10$, and $n_{\text{DMD}} = 90$. Smaller relative loss means better performance.

VQE from the optimal point marked by red points rather than the last DMD prediction. In Figure 5d, we show the MLP-SW-DMD predictions of Adam as another example. The DMD predictions do not exactly align with the full VQE, but they still successfully lower the energy in general. When the energy is close to convergence, the DMD predictions may lead to minor energy increase, but this does not affect the final performance.

## F  RELATIVE LOSS FOR THE ISING MODEL

Table 1 demonstrates numerically the results from Figure 3c.

| Method | Relative Gradient Steps | | | | | | |
| --- | --- | --- | --- | --- | --- | --- | --- |
|  | 7% | 8% | 10% | 15% | 20% | 25% | 30% |
| Pure VQE | 56.3% | 52.2% | 44.6% | 29.2% | 18.4% | 11.0% | 5.8% |
| DMD | 41.2% | 9.2% | 32.1% | 15.1% | 11.0% | 5.7% | 1.9% |
| SW-DMD | **4.3%** | **4.1%** | 5.7% | 1.4% | 0.5% | 0.2% | 0.004% |
| MLP-DMD | 18.8% | 12.0% | 10.2% | 9.9% | 4.1% | 0.5% | 0.09% |
| MLP-SW-DMD | 6.8% | 6.6% | **3.8%** | **0.2%** | **0.04%** | **0.007%** | **0.003%** |
| CNN-DMD | 11.8% | 6.9% | 5.2% | 6.3% | 3.0% | 2.3% | 0.9% |

Table 1: Relative loss (in %) as a function of the method used and the relative gradient steps (in %). Lower relative loss is better. Our methods significantly improve the standard DMD.

The relative loss has a trend of decrease when the relative gradient steps increase. However, this trend is not strictly monotonic, because using a longer history of optimization (more relative gradient steps) may occasionally lead to worse prediction. Because of the complicated space dynamics, it may be challenging to fit a longer time series of parameters.

## G  EFFECTS ON VARYING $n_{\text{SW}}$

We perform an ablation study for the effects on the performance from varying $n_{\text{SW}}$ in SW-DMD. We use the quantum Ising model at $h = 0.5$ on 12 qubits with Adam and choose $n_{\text{sim}} = 10$ and $1 \leq n_{\text{SW}} \leq 9$ ($n_{\text{SW}} < n_{\text{sim}}$ is a strict requirement). We use $n_{\text{total}} = 300$ steps of full VQE and $n_{\text{DMD}} = 90$.

Figure 6 shows the relative loss versus $n_{\text{SW}}$ with $10\%$ and $20\%$ relative gradient steps defined in Sec. 5.1.2. When the relative gradient steps are $10\%$, the relative loss is the highest when $n_{\text{SW}} = 1$ which is equivalent to the standard DMD. This shows that SW-DMD in general has a better performance than the standard DMD. The relative loss is low in the intermediate regime of $n_{\text{SW}}$ near

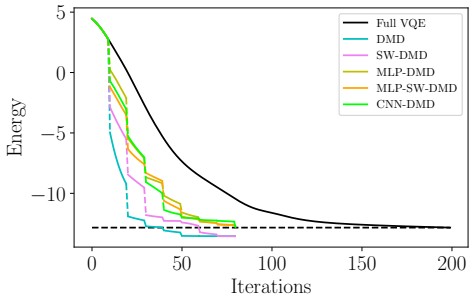

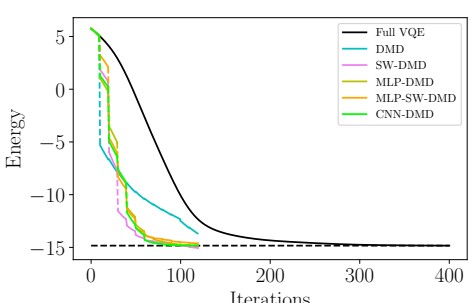

(a) Natural gradient 10-qubit results with $n_{sim} = 10$, $n_{DMD} = 40$, and $n_{SW} = 6$ for SW-DMD and MLP-SW-DMD. For the various DMD methods, the solid piecewise curves are actual gradient steps, and the dashed lines connecting them indicate when the DMD prediction is applied.

(b) Adam 12-qubit results with $n_{sim} = 10$, $n_{DMD} = 40$, $n_{SW} = 6$ for SW-DMD and MLP-SW-DMD. For the various DMD methods, solid parts are VQE runs, and the dashed parts are DMD predictions.

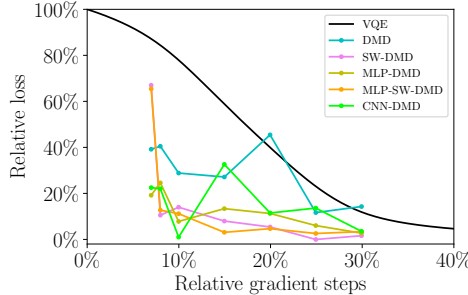

(c) Adam 12-qubit results for relative loss versus relative gradient steps. Less relative gradient steps mean less quantum resource used. Lower relative loss indicates better performance of the optimization.

Figure 7: Experimental results for Heisenberg model at $J_z = 0.5$.

$n_{SW} = 4$ but is even lower at a large window size $n_{SW} = 9$. However, the best performance achieved by the maximum $n_{SW}$ is not universal. As we increase the relative gradient steps from 10% to 20%, at each $n_{SW}$, the relative loss decreases, but $n_{SW} = 9$ no longer has the lowest relative loss compared to the intermediate window size near $n_{SW} = 5$.

Theoretically, the performance versus $n_{SW}$ can be related to the shape of the extended data matrix in Eq. 7 where $n_{SW} = d + 1$. The number of rows is $n_{SW} \cdot n_{params}$, and the number of columns is $n_{sim} - n_{SW} + 1$. There can be a trade-off in choosing $n_{SW}$. Increasing $n_{SW}$ may enhance the ability of the time-delay embedding to capture the nonlinear dynamics, but on the other hand, this may also lead to fewer columns which potentially means a less independent information. Heuristically, an intermediate window size $n_{SW}/n_{params} \sim 0.5$ may have a good performance, but this is not a strict criterion. In some cases, we find empirically that $n_{SW}/n_{params} > 0.5$ can have a better performance, such as $n_{SW}/n_{params} = 0.9$ with 10% relative gradient steps in Figure 6.

## H    RESULTS OF THE QUANTUM HEISENBERG MODEL

Similar to the experiments we have on the quantum Ising model in Figure 3 a,b,c, we implement the experiments for the 12-qubit quantum Heisenberg model at $J_z = 0.5$ with all the hyperparameters the same as the Ising model experiments.

Figure 7a shows the results from the 10-qubit noiseless natural gradient simulations. All the DMD methods are able to significantly accelerate the quantum optimization. DMD and SW-DMD even lower the energy lower than the final full VQE simulation. This is maybe because the predictions by

| Method | Relative Gradient Steps | | | | | | |
|--------|------|------|------|------|------|--------|------|
|        | 7%   | 8%   | 10%  | 15%  | 20%  | 25%    | 30%  |
| Pure VQE | 87.2% | 84.4% | 77.9% | 58.8% | 40.0% | 23.1% | 11.9% |
| DMD | 39.2% | 40.5% | 28.8% | 27.1% | 45.4% | 11.7% | 14.3% |
| SW-DMD | 67.0% | **10.5%** | 14.0% | 8.0% | 5.4% | **-0.03%** | **1.6%** |
| MLP-DMD | **19.2%** | 24.6% | 7.8% | 13.4% | 11.3% | 6.0% | 2.7% |
| MLP-SW-DMD | 65.4% | 12.7% | 11.2% | **3.1%** | **4.7%** | 2.6% | 3.4% |
| CNN-DMD | 22.5% | 22.0% | **1.0%** | 32.6% | 11.5% | 13.6% | 3.5% |

Table 2: Relative loss (in %) from the quantum Heisenberg model at $J_z = 0.5$ with Adam simulations as a function of the method used and the relative gradient steps (in %). Lower relative loss is better. Our methods significantly improve the standard DMD.

DMD and SW-DMD find a different trajectory from the full VQE trajectory, which takes $\boldsymbol{\theta}$ out of a local minimum to a lower energy.

Figure 7b shows the results from the 12-qubit noiseless Adam simulations. All the DMD methods are able to significantly accelerate the quantum optimization. With the fixed number of iterations, the standard DMD method successfully lowers the energy but still does not reach the final loss of full VQE.

Figure 7 shows the results of the ablation study from 12-qubit noiseless Adam simulations, with the same results presented in Table 2 for better numerical resolution. We use $n_{\mathrm{SW}} = 6$ for SW-DMD and MLP-SW-DMD. Relative loss and relative gradient steps are defined in Sec. 5.1.2. Almost all points from various DMD methods are below the pure VQE curve, which indicate successful acceleration. The only exception occurs at $20\%$ ralative gradient steps for the standard DMD method, where the relative loss from the standard DMD is a little higher than the pure VQE. When we start the follow-up piece of VQE from the DMD optimal prediction (with an energy lower than the last iteration of the previous piece of VQE), the DMD optimal prediction can still be at a worse position for optimization than the pure VQE in the space of parameters, because the energy landscape can be complicated.

In Table 1, the best performances among the various DMD methods are achieved by SW-DMD and MLP-SW-DMD, and this also holds in general in Table 2. However, here in Table 2, at $7\%$ relative gradient steps, neither SW-DMD nor MLP-SW-DMD have the best performance. This can be because we $n_{\mathrm{sim}} = 7$ in this case, which leads to the number of columns in the time-delay embedding is only $n_{\mathrm{sim}} - n_{\mathrm{SW}} + 1 = 2$, the minimum number for constructing the dynamics. As is analyzed in Appendix E, a small number of columns can sometimes lead to a relatively bad performance. At $8\%$ relative gradient steps, CNN-DMD has the best performance compared to other methods, which indicates the potential power of CNN-DMD if the CNN training can be properly done.

## I    RESULTS AT MORE VALUES OF $h$

We show the 12-qubit noiseless Adam results at $h = 0.9, 2.0$ in Figure 8 with the same setup as we do for $h = 0.5$ in Sec. 5.1.1. All the DMD methods significantly accelerate the quantum optimization, although some of them in Figure 8b may still need more iterations to get closer to the minimum loss of full VQE. In Figure 8a, the SW-DMD and MLP-SW-DMD find even lower energies than full VQE, similar to what we observe in Figure 7a.

## J    NOISY QUANTUM SIMULATIONS

To study the problem with a more realistic setup, we also perform simulations using a noise model FakeLima, which mimics a real IBM quantum computer Lima, on a 5-qubit Ising model with $h = 0.5$ with the periodic boundary condition. The optimizer is Adam with learning rate $0.01$. We use

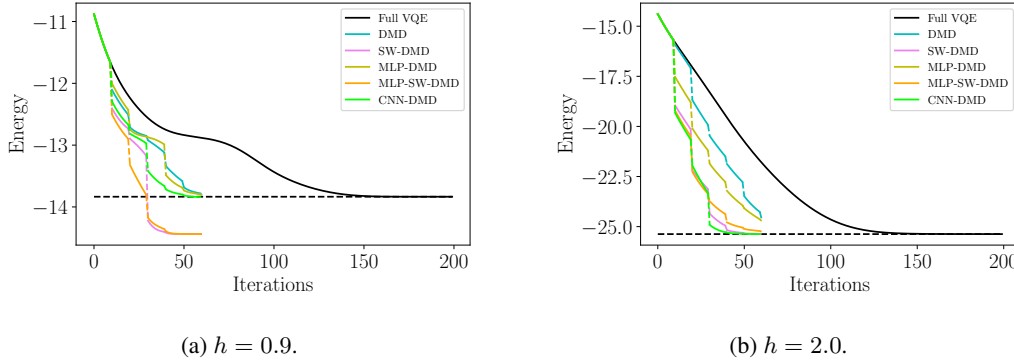

(a) $h = 0.9$.

(b) $h = 2.0$.

Figure 8: 12-qubit noiseless Adam simulation results at (a) $h = 0.9$ and (b) $h = 2.0$ with $n_{\text{sim}} = 10$, $n_{\text{DMD}} = 90$, $n_{\text{total}} = 200$. We use $n_{\text{SW}} = 6$ for SW-DMD and MLP-SW-DMD. For the various DMD methods, the solid piecewise curves are actual gradient steps, and the dashed lines connecting them indicate when the DMD prediction is applied.

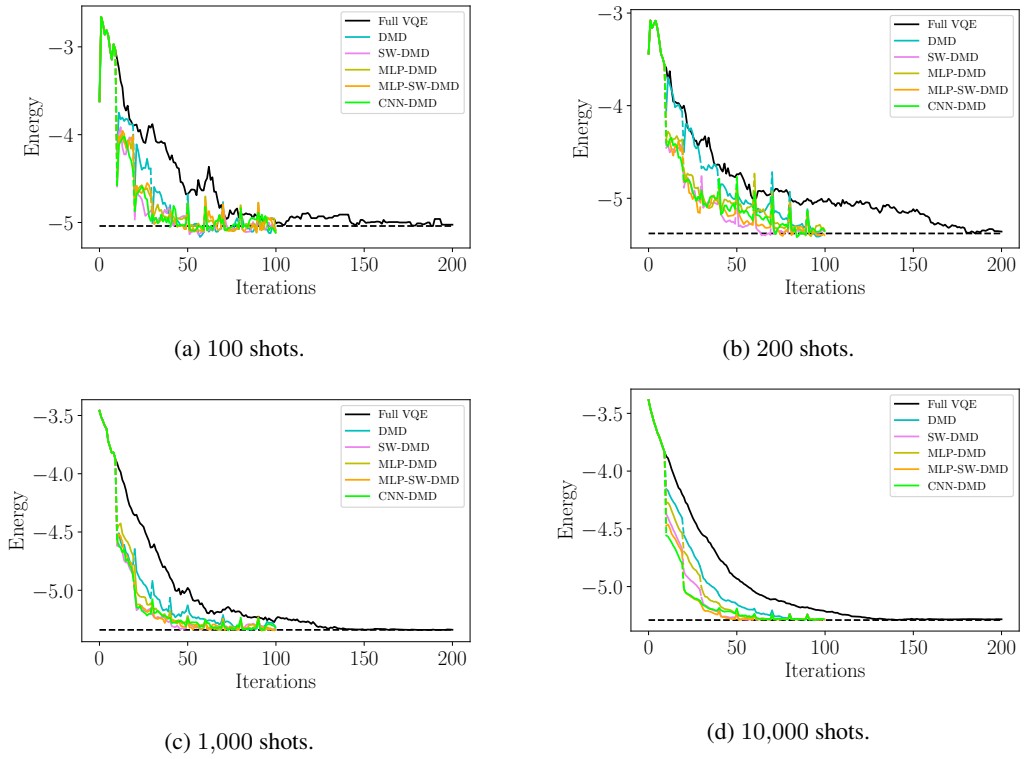

(a) 100 shots.

(b) 200 shots.

(c) 1,000 shots.

(d) 10,000 shots.

Figure 9: Simulations on the noise model FakeLima with different numbers of shots for quantum measurement: (a) 100 shots (b) 200 shots (c) 1,000 shots (d) 10,000 shots. Optimization histories for energy of full VQE and various DMD methods are shown. Dashed lines indicate that the DMD methods are used to accelerate the optimization.

the RealAmplitudes ansatz with 2 layers (10 parameters) with circular entanglement. We have also applied measurement error mitigation (Barron & Wood, 2020) to reduce the effect of quantum noise.

The evaluation of loss on the quantum computer is based on quantum measurements. With more shots of measurements, the statistical error will be reduced, but the needed quantum resources will increase. In practice, it is interesting to see whether our algorithm works with a relatively small number of shots, as we show below.

In Fig. 9, we show the simulated optimization histories of the full VQE and various DMD methods. We run the full VQE for $n_{\text{total}} = 200$ iterations. We choose $n_{\text{sim}} = 10$ and $n_{\text{DMD}} = 20$. SW-DMD and MLP-SW-DMD use window size $n_{\text{SW}} = 6$. Both the loss function accuracy and the gradient precision scale with the number of shots $n_{\text{shots}}$ as $O(1/\sqrt{n_{\text{shots}}})$. Even with a very small number of shots such as $n_{\text{shots}} = 100$, all the DMD methods can accelerate the quantum optimization, as the loss with DMD decreases quicker than the full VQE. The final energy with 100 shots is not as low as the final energy from more shots, but this is not because of any inferiority of the DMD methods, since the full VQE also has difficulty to reach a lower energy. Our noisy simulations demonstrate that our methods are promising in cases closer to a realistic setup.

The spikes in the DMD results occur every $n_{\text{sim}} = 10$ iterations at the beginning of every piece of VQE, and are more distinct when the shots are smaller. This may be because it is harder to make predictions of the VQE history when the parameter updates time series from measurement is more noisy. Despite the occasional spikes in energy, the DMD methods still help to get closer to the better parameters for optimization in later steps.

## K    ADDITIONAL DISCUSSION ON REAL LIMA EXPERIMENTS

### K.1    CHOICE OF OPTIMIZER AND HYPERPARAMETERS

We use SPSA instead of Adam, because the required quantum resource for a purely gradient-based optimizer such as Adam is too much at this stage due to our limited access to quantum computers. SPSA is quasi-graident-based and more realistic to implement on a real quantum computer. With the SPSA optimizer, every iteration of update takes 1 center measurement and 2 gradient measurements in only one random direction of the parameter space, so each iteration takes 3 measurements. The smaller number of gradient measurements makes it easier to implement on near-term real quantum computers, compared to Adam. However, using SPSA may also require a higher number of iterations. The DMD method only takes 1 center measurement without gradient measurements. The gradient-based method should have more advantage than SPSA as the number of parameters increases, since SPSA provides much more noisy gradient estimates. There is no fundamental limitation to implement pure gradient-based methods, but it is more time and resource consuming given the limited accessibility of quantum computers. Indeed, one of our main motivations to develop the Koopman operator learning algorithm is to reduce the cost for gradient-based optimization. Since the number of parameters in quantum circuit usually scales with the number of qubits in practice, our approach should be able to demonstrate more gains for large-scale quantum optimization with gradient-based methods.

To mitigate the randomness in the direction of gradient measurements in SPSA, we choose $n_{\text{sim}} = 20$ for $n_{\text{params}} = 10$. Within these first 20 VQE simulations, each parameter on average has one forward step gradient measurement and one backward step gradient measurement. We also observe that a relatively large sliding window may help the learning. We choose $n_{\text{SW}} = 15$ for SW-DMD and MLP-SW-DMD, and $n_{\text{sim}} = 20$ instead of 10 so that $n_{\text{SW}}/n_{\text{sim}}$ is relatively close to 1. The standard DMD, MLP-DMD, CNN-DMD already have unstable performances with the same setups on FakeLima. This probably comes from the fact that SPSA with quantum noise returns very noisy parameter updates compared to the true gradient-based method, since it makes random perturbations in the parameter space, and the quantum noise further biases the measurement. This might make the Koopman operator learning much more challenging.

### K.2    NOISE ANALYSIS

The noisy decay of the energy in the curve in Figure 3d could be related to multiple sources: (1) quantum noise, (2) experimental instability in the real lab apparatus, such as day-to-day calibration, (3) statistical error from quantum measurements, (4) randomness from SPSA. (3) is not a dominant factor here, since the statistical errors are relatively small in our experiments. To help probe and diagnose these various effects, we also perform the same experiments with the same hyperparameters on FakeLima shown in Appendix K.3. Both SW-DMD and MLP-SW-DMD can successfully accelerate the quantum optimization on FakeLima. This might indicate that there is in fact a difference between the real Lima and FakeLima, since the real Lima is subject to daily fluctuation while Fake-

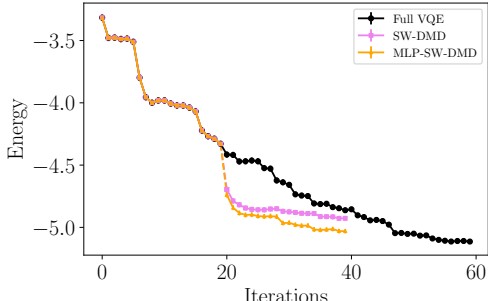

Figure 10: FakeLima results.

| Method | Relative Gradient Steps | | | | |
|--------|------|------|------|------|------|
| | 10% | 20% | 30% | 40% | 50% |
| Pure QML | 81.8% | 92.8% | 96.0% | 95.8% | 95.6% |
| DMD | **95.2%** | 98.4% | 98.8% | 98.8% | **99.2%** |
| SW-DMD | **95.2%** | **98.8%** | **99.0%** | **99.2%** | 99.0% |
| MLP-DMD | 93.6% | 96.4% | 97.6% | 98.4% | 98.6% |
| MLP-SW-DMD | 94.0% | 98.6% | **99.0%** | **99.2%** | 99.0% |
| CNN-DMD | 89.2% | 95.4% | 97.4% | 98.6% | 98.4% |

Table 3: Test accuracy (in %) from quantum machine learning in Sec. 5.2 on the filtered MNIST dataset as a function of the method used and the relative gradient steps (in %).

Lima is a fixed noise model. However, (1) and (4) may still be possible to account for the relatively insignificant acceleration of MLP-SW-DMD on real IBM Lima.

### K.3 COMPARISON BETWEEN REAL LIMA AND FAKELIMA

We perform simulation using FakeLima on the same 5-qubit Ising model at $h = 0.5$ with SPSA, as we do on the real Lima in Sec. 5.1.3 and Figure 3d. The FakeLima results are shown in Figure 10. Both SW-DMD and MLP-SW-DMD significantly accelerate the quantum optimization on FakeLima. The performance in the FakeLima case is better than in the real Lima case, and the diagnosis and analysis are detailed in Sec. 5.1.3.

## L  QML ARCHITECTURE AND TRAINING DETAILS

In our QML example in Sec. 5.2, the quantum computer has $N = 10$ qubits. The interleaved encoding gate consists of 9 layers. Each layer has a rotational layer and a linear entanglement layer. On the rotational layer, each qubit has three rotational gates $R_X, R_Z, R_X$ in a sequence, where

$$R_X(\theta) = \exp\left(-i\theta X/2\right) = \begin{bmatrix} \cos\left(\theta/2\right) & -i\sin\left(\theta/2\right) \\ -i\sin\left(\theta/2\right) & \cos\left(\theta/2\right) \end{bmatrix}, \tag{15}$$

$$R_Z(\theta) = \exp\left(-i\theta Z/2\right) = \begin{bmatrix} \exp\left(-i\theta/2\right) & 0 \\ 0 & \exp\left(i\theta/2\right) \end{bmatrix}. \tag{16}$$

Therefore, each layer has 30 rotational angles, and the whole QML architecture has 270 rotational angles, $i.e.$, $n_{\mathrm{params}} = 270$. Since each input example $x$ is 256-dimensional, we only use the first 256 rotational angles to encode the input data. The parameters $\theta$ are also encoded in the rotational angles such that the angles are $x + \theta$.

Quantum measurements, as the last part of the quantum circuit, map the quantum output to the classical probability data. In each quantum measurement, each qubit is in the 0-state or the 1-state,

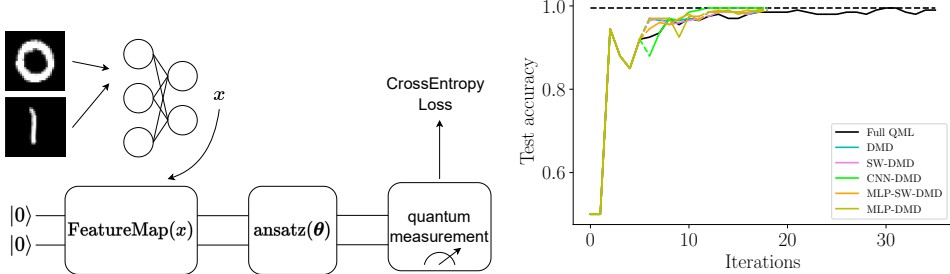

(a) Hybrid classical-quantum machine learning architecture.

(b) Accuracy of binary classification for full QML simulation and DMD methods. For the various DMD methods, solid parts are QML runs, and the dashed parts are DMD predictions.

Figure 11: Hybrid classical-quantum machine learning architecture and results.

and the probability for the qubit to be in the 0-state is between 0 and 1. We regard this probability as the probablity for the image to be a digit "1". We only use the probability on the 5th qubit ($i = 5$) as the output and compute the cross-entropy loss with the labels.

In the phase of training, all the training examples share the same $\boldsymbol{\theta}$ but have different $\boldsymbol{x}$, and we optimize the final average loss with respect to $\boldsymbol{\theta}$ as

$$\boldsymbol{\theta}^* = \arg\min_{\boldsymbol{\theta}} \frac{1}{n_{\text{train}}} \sum_{i=1}^{n_{\text{train}}} \mathcal{L}(\boldsymbol{x}_i^{\text{train}}; \boldsymbol{\theta}), \tag{17}$$

in the case of $n_{\text{train}}$ training examples. The combination $\boldsymbol{x}_i^{\text{train}} + \boldsymbol{\theta}$ is entered into the quantum circuit rotational angles, and we need to build $n_{\text{train}}$ separate quantum circuits (which can be built either sequentially or in parallel). The interleaved encoding gate as a whole can be viewed as a deep layer, and serve dual roles of encoding the classical data $\boldsymbol{x}$ and containing the machine learning parameters $\boldsymbol{\theta}$. In the phase of inference, with the optimal parameter $\boldsymbol{\theta}^*$, we feed each test example $\boldsymbol{x}_i^{\text{test}}$ into the quantum circuit as a combination $\boldsymbol{x}_i^{\text{test}} + \boldsymbol{\theta}^*$ and measure the output probability to compute the test accuracy.

We use 500 training examples and 500 test examples. During training, we use the stochastic gradient descent optimizer with the batch size 50 and learning rate 0.05. The full QML training has $n_{\text{total}} = 400$ iterations. We choose $n_{\text{sim}} = 10$, $n_{\text{DMD}} = 20$, and $n_{\text{SW}} = 6$ for SW-DMD and MLP-SW-DMD.

In neural DMD including MLP-DMD, MLP-SW-DMD, CNN-DMD, we use layerwise partitioning that groups $\boldsymbol{\theta}$ by layers in the encoding gate. There are 9 groups with group size 30. We perform neural DMD for each group, so that the number of parameters in the neural networks for DMD is not too large. After predicting each group separately, we combine all groups of $\boldsymbol{\theta}$ to evaluate the loss and accuracy. In CNN-DMD for QML, we use 80k steps for sufficient CNN training.

The numerical values in Figure 4b of test accuracy from various methods at relative gradient steps compared to pure QML are given in Table 3. All the DMD methods significantly accelerate the QML training.

# M  HYBRID CLASSICAL-QUANTUM MACHINE LEARNING

We illustrate another example of Koopman operator learning for QML using a hybrid quantum-classical neural network with its architecture shown in Figure 11a. We consider supervised learning for binary classification over a filtered MNIST dataset with digits 0 and 1. First the MNIST data is fed into a classical neural network for the feature $x$. Then, the feature $x$ is encoded into the quantum computer via a sequence of quantum gates for feature mapping. Then, the feature-encoded quantum state goes through an ansatz layer with parameter $\boldsymbol{\theta}$, similar to the case of quantum optimization.

We use the results from quantum measurements as the output of the hybrid neural network. We test our Koopman operator learning algorithms with the above architecture. The full QML training has $n_{\text{total}} = 36$ iterations. We choose $n_{\text{sim}} = 6$, $n_{\text{DMD}} = 6$, and $n_{\text{SW}} = 3$ for SW-DMD and MLP-SW-DMD. Figure 11b shows that DMD methods can achieve reasonably good accuracy while using less quantum resource than the full QML.

The classical neural network in the hybrid architecture consists of [Conv2D(in-channel=1, out-channel=2, kernel-size=5), ReLU(), MaxPool2D(size=2), Conv2D(in-channel=2, out-channel=2, kernel-size=5), ReLU(), Linear(in-dim=2, out-dim=2), ReLU(), Linear(in-dim=2, out-dim=2)].

The feature mapping is realized through the ZZFeatureMap on two qubits with one repetition from Qiskit. The ansatz after the feature mapping is the RealAmplitude ansatz with one repetition. The quantum measurements are performed on the $Z$-basis of two qubits, which return the expectation value of $\langle Z_1 Z_2 \rangle$ as an output $y$. Then, $y$ and $1-y$ are used as logits for the CrossEntropy loss. Adam is used as the optimizer with learning rate $5 \times 10^{-4}$. 50 training examples and 100 test examples are used for each digit.

**Summary of the functionality of the different components in Figure 11a.** In the beginning a classical neural network takes the original input of image data to learn the representation $x$ in a classical feature space. Then the rest of components in the architecture is quantum. The features $x$ are fed into the FeatureMap($x$) quantum gate as a layer to encode the classical data into quantum data. Then, the ansatz($\boldsymbol{\theta}$) quantum gate as a follow-up layer serves the same role as the ansatz layer in the case of quantum optimization. Finally, the quantum measurement takes the output quantum data into classical outputs.

