# OpenReview forum: "Koopman Operator Learning for Accelerating Quantum Optimization and Machine Learning"
_ICLR.cc/2023/Conference — Submitted to ICLR 2023_

### Official Review · Reviewer_zjTr · 2022-10-20

**Confidence:** 2
**Correctness:** 3
**Technical Novelty And Significance:** 3
**Empirical Novelty And Significance:** 2
**Recommendation:** 6

**Clarity, Quality, Novelty And Reproducibility:**

Clarity of the paper is good.

The main novelty should be finding a good application in quantum computing. Sliding window method offers some novelty, but to my ignorant, I do not know of any other similar approach.


**Strength And Weaknesses:**

The motivation of the paper is good. It is one plausible solution to an important problem. The method of Koopman operator learning using sliding window and neural network to learn dictionary, can potentially be use for optimising other classical computing algorithm. its advantage is most in quantum computing.

I have not thought through thoroughly if using the sliding window approach is theoretically sound. Certainly did not sit down, write the required equations and convince myself this is a theoretically sound approach. Loss function of Eq. (8) (or similar) can be constructed in many ways, sliding window way etc, seems arbitrary to me. I understand several other works in the area of Koopman operator also uses similar approach of constructing some measurable / observable and then do optimization. In the limit of infinite observables, one can agree that the trained Koopman operator can represent the underlying dynamics accurately. There is also an issue of how to invert the "g" function to get back the underlying "x" (Eq. (1)). In this paper, it seems g is the identity function.




**Summary Of The Paper:**

The paper propose using Koopman operator to speed up the optimization of hybrid quantum computing algorithm. The motivation is gradient step in quantum computer takes many forward evaluations (linear in number of parameters). Using Koopman operator I the classical computing sense, to predict future optimal quantum machine learning parameters. then pick the best, optimal time, and then do more predictions from there.


**Summary Of The Review:**

Good motivation for this work. Not absolutely sure if the equations are theoretically sound due to my ignorant.
My sense is this paper is in between (6) Marginally above acceptance threshold and (8) accept, good paper.

---

> ### Author Response · Authors · 2022-11-14
> **References**
>
> [1] Daniel Dylewsky, Eurika Kaiser, Steven L Brunton, and J Nathan Kutz. Principal component trajectories for modeling spectrally continuous dynamics as forced linear systems. Physical Review E, 105(1):015312, 2022.
>
> [2] Brunton, Bingni W., Lise A. Johnson, Jeffrey G. Ojemann, and J. Nathan Kutz. "Extracting spatial–temporal coherent patterns in large-scale neural recordings using dynamic mode decomposition." Journal of neuroscience methods 258 (2016): 1-15.
>
> [3] Takens, F. (1981). Detecting strange attractors in turbulence. In: Rand, D., Young, LS. (eds) Dynamical Systems and Turbulence, Warwick 1980. Lecture Notes in Mathematics, vol 898. Springer, Berlin, Heidelberg. https://doi.org/10.1007/BFb0091924

---

> ### Author Response · Authors · 2022-11-14
> **Clarification of Algorithms Details and Summary of Theoretical, Algorithmic and Experimental Contributions**
>
> > There is also an issue of how to invert the "g" function to get back the underlying "x" (Eq. (1)). In this paper, it seems g is the identity function.
>
> We want to clarify that
> * For the standard DMD, g is the identity function
> * For SW-DMD and neural DMD, g is not the identity function, but comes from time-delay embedding and the neural encoder. K in our design of SW-DMD and neural DMD can be nonsquare, and it directly connects the embedded earlier data g(x(t)) to the unembedded later data x(t+1) such that x(t+1) = K g(x(t)). No explicit inversion of g is needed as long as the parameter prediction can be uniquely determined, since the operator (K g) can act on x(t) repeatedly to get a prediction, e.g., in k steps forward, x(t+k+1) = (K g)^k (x(t+1)). We instantiate the discussion of nonsquare K in the more concrete context of our optimization problems, in Sec. 4.2 and the last paragraph of Appendix C, where g is Phi for SW-DMD and Phi_alpha for neural DMD.
>
> > I have not thought through thoroughly if using the sliding window approach is theoretically sound. Certainly did not sit down, write the required equations and convince myself this is a theoretically sound approach. Loss function of Eq. (8) (or similar) can be constructed in many ways, sliding window way etc, seems arbitrary to me.
>
> We would like to clarify that
> * The sliding window method is inspired by the time-delay embedding Hankel method [1] and the time-delay augmented DMD [2] which we cite in the paper and it can be justified by the Taken’s time-delay embedding theorem [3].
> * Though our sliding windows DMD also uses the time-delay embedding, it is different from the Hankel [1] approach and the time-delay augmented DMD [2]. Unlike the Hankel approach that performs SVD, we design the approach to match the time-delay embedding with a rectangle Koopman matrix directly, which is more efficient for our quantum optimization and quantum machine learning tasks. In addition, we find that the Hankel approach is not stable for quantum optimization applications, while our SW-DMD is more robust. While the time-delay augmented DMD uses time-delay embedding as a data augmentation, it produces a square Koopman operator and may have the difficulty to uniquely determine the prediction in the case of noisy parameter dynamics in optimization tasks on quantum computers. Our approach can uniquely determine the prediction and we also find that it has more stable performance than the time-delay augmented DMD.
> * The loss function of Eq. (8) is designed to match the time-delay embedding between two near time steps. The Frobenius norm is used such that it can be turned into a least square problem and solved efficiently with linear algebra techniques.
>
> > The main novelty should be finding a good application in quantum computing. Sliding window method offers some novelty, but to my ignorant, I do not know of any other similar approach.
>
> With the general response above, we would like to point out that our contributions come from the following perspectives
> * Theoretically, the gradient optimization scales with the number of parameters and is much more costly on quantum computers compared to classical computers. We address the issue by utilizing Koopman operator learning. We also connect the Koopman operator theory to the quantum natural gradient, which does not exist in the classical machine learning setup and builds a well-motivated theoretical foundation for our work.
> * Algorithmically, we develop SW-DMD and neural DMD with an iterative optimization protocol tailored for quantum optimization and quantum machine learning. The previous classical work only uses standard DMD with no follow-up optimization after the DMD prediction using a large amount of data. However, we notice that this could not be applied to quantum optimization and quantum machine learning with much fewer training data. Hence, our methods can be efficiently used with few training data, accelerate the gradient method, and be robust to long-time prediction error and noise. The algorithmic development is new and important to the field.
> * Experimentally, we design and implement a series of systematic experiments over various models with different optimizers for both quantum optimization and quantum machine learning. We have further applied our approach to a real IBM quantum computer and observed acceleration in realistic setups. All the experiments successfully demonstrate that our approach provides a powerful tool for accelerating quantum optimization and quantum machine learning.

---

> ### Author Response · Authors · 2022-11-17
> **Looking forward to your feedback**
>
> Dear Reviewer zjTr,
>
> We would be grateful if you can confirm if our response has addressed your concerns and let us know if there is any other question or suggestion. In the following, we summarize the key points of our response:
> * We explain our distinct contributions beyond the previous literature. Theoretically our approach addresses the gradient complexity challenges on quantum computers and connects the Koopman operator theory with quantum natural gradient, both of which do not exist in classical machine learning. Algorithmically we develop new robust iterative DMD algorithms tailored for efficient acceleration of quantum optimization and quantum machine learning.
> * We perform a new series of systematic experiments on various systems of large sizes for both quantum optimization and quantum machine learning, through simulations and real quantum computers.The new results strongly demonstrate the success of our methods.
> * We improve the presentation and discussion of figures, tables and results, which clarifies the confusions in the previous manuscript and makes the paper more comprehensive.
>
> We look forward to hearing your feedback!

---

> ### Author Response · Authors · 2022-12-05
> **Has our response addressed your concerns?**
>
> Dear Reviewer,
>
> Thank you so much for your constructive review which allowed us to improve our paper! In our response, we highlighted the theoretical and algorithmic novelty of our contributions, performed in-depth new experiments, and improved the presentation of the paper. We believe we were able to respond in depth to all of your concerns and kindly ask you to consider increasing your score.
>
> Best wishes,
> The authors

---

### Official Review · Reviewer_TgmW · 2022-10-24

**Confidence:** 4
**Correctness:** 3
**Technical Novelty And Significance:** 1
**Empirical Novelty And Significance:** 1
**Recommendation:** 3

**Clarity, Quality, Novelty And Reproducibility:**

Overall, the paper is very clear and can be probably reproduced. Based on the discussion above, I believe the novelty of the paper is somewhat limited.

**Strength And Weaknesses:**

In my opinion, the strongest point of the paper is the application domain of quantum machine learning (QML) and the incorporation of Koopman-based methodologies in it. Having said that, it should be noted that the technical problem solved in this paper is essentially the ability to approximate complex dynamics with Koopman methods, and the relation to QML is only by application.

Unfortunately, there are several weak points. First, the same observation and application was already considered in Dogra et al. (and other papers, some of which cited in this paper). In this context, the main difference between this paper and the paper of Dogra et al. is in the evaluation on QML tasks vs. vision tasks. While this difference may be significant in certain applications and algorithms, I do not believe this is the case here, as they essentially optimize a standard neural network, similar to those considered in Dogra et al.

Second, the proposed approaches to approximate the Koopman operator generally ignore many similar if not identical ideas in the literature. Both sliding windows with time delay coordinates and neural DMD have been heavily studied before. Properly discussing these approaches and positioning the proposed method in this context is essential.

**Summary Of The Paper:**

This paper proposes to model the variational quantum eigensolver optimization as a dynamical system, and approximate its iterative updates via Koopman-based approaches. In particular, the authors propose a sliding window DMD method, and a neural DMD approach. The authors evaluate their approach on quantum machine learning tasks, and compare it to standard DMD.

**Summary Of The Review:**

In summary, while the application domain of this paper is interesting and refreshing, it is essentially a different application of an existing paper. Moreover, the proposed methods are similar or equivalent to existing work in the field. I suggest the authors to conduct a proper literature search in Koopman and DMD communities (see e.g., Hankel DMD and Koopman Autoencoder), and to position their contribution accordingly. Due to these reasons, I think the paper should not be published in its current form.

---

> ### Author Response · Authors · 2022-11-14
> **References**
>
> [1] Harrigan, M.P., Sung, K.J., Neeley, M. et al. Quantum approximate optimization of non-planar graph problems on a planar superconducting processor. Nat. Phys. 17, 332–336 (2021). https://doi.org/10.1038/s41567-020-01105-y.
>
> [2] S. Ebadi, A. Keesling, M. Cain, T. T. Wang, H. Levine, D. Bluvstein, G. Semeghini, A. Omran, J.-G. Liu, R. Samajdar, X.-Z. Luo, B. Nash, X. Gao, B. Barak, E. Farhi, S. Sachdev, N. Gemelke, L. Zhou, S. Choi, H. Pichler, S.-T. Wang, M. Greiner, V. Vuleti ́c , and M. D. Lukin. Quantum optimization of maximum independent set using rydberg atom arrays. Science, 376(6598): 1209–1215, jun 2022. doi: 10.1126/science.abo6587. URL https://doi.org/10.1126% 2Fscience.abo6587.
>
> [3] Akshunna S Dogra and William Redman. Optimizing neural networks via koopman operator theory. Advances in Neural Information Processing Systems, 33:2087–2097, 2020.
>
> [4] Daniel Dylewsky, Eurika Kaiser, Steven L Brunton, and J Nathan Kutz. Principal component trajectories for modeling spectrally continuous dynamics as forced linear systems. Physical Review E, 105(1):015312, 2022.
>
> [5] Brunton, Bingni W., Lise A. Johnson, Jeffrey G. Ojemann, and J. Nathan Kutz. "Extracting spatial–temporal coherent patterns in large-scale neural recordings using dynamic mode decomposition." Journal of neuroscience methods 258 (2016): 1-15.
>
> [6] Bethany Lusch, J. Nathan Kutz, and Steven L. Brunton. Deep learning for universal linear embeddings of nonlinear dynamics. Nature Communications, 9(1), nov 2018. doi: 10.1038/
> s41467-018-07210-0. URL https://doi.org/10.1038%2Fs41467-018-07210-0.
>
> [7] Omri Azencot, N Benjamin Erichson, Vanessa Lin, and Michael Mahoney. Forecasting sequential data using consistent koopman autoencoders. In International Conference on Machine Learning, pp. 475–485. PMLR, 2020.

---

> > ### Comment · Reviewer_TgmW · 2022-11-23
> > **response to authors**
> >
> > I acknowledge that I read the authors' responses.

---

> > > ### Author Response · Authors · 2022-11-23
> > > **Looking forward to your feedback on our responses**
> > >
> > > Dear Reviewer,
> > >
> > > Thank you for reading our response. We will appreciate it to know how our responses help alleviate your concerns and update your evaluation. Please just kindly let us know if there is any suggestion, and we look forward to a constructive discussion. Thanks!

---

> ### Author Response · Authors · 2022-11-14
> **Our Work Develops New Algorithms Tailored For Efficient Acceleration of Quantum Optimization and Quantum Machine Learning**
>
> > Second, the proposed approaches to approximate the Koopman operator generally ignore many similar if not identical ideas in the literature. Both sliding windows with time delay coordinates and neural DMD have been heavily studied before. Properly discussing these approaches and positioning the proposed method in this context is essential. I suggest the authors to conduct a proper literature search in Koopman and DMD communities (see e.g., Hankel DMD and Koopman Autoencoder), and to position their contribution accordingly.
>
> We have added more relevant literature in both the related work and the relevant contexts. While there are sliding windows with time delay coordinates and neural DMD for time series prediction, we would like to point out several technical distinctions of our approach.
> * Our focus is on optimization instead of dynamics prediction. Our algorithm is not the simple DMD but various new DMD with an iterative optimization protocol for acceleration. Since gradient optimization is more challenging in quantum computation compared to classical optimization, we notice that the one-shot prediction approach (with no follow-up optimization after the DMD prediction) with training data, consisting of many parameters, in the previous work [3] does not work well for accelerating quantum optimization and machine learning. Hence, we develop the iterative DMD optimization protocol shown in Fig. 1 in the manuscript. Due to the complexity separation between gradient computation and prediction verification on quantum computers, we can verify our prediction of parameters for many step updates efficiently and choose the optimal one. This algorithm can effectively accelerate the optimization with a few training data records and is robust against long-time prediction error and noise, which is crucial for realistic quantum optimization and machine learning.
> * Though our sliding windows DMD also uses the time-delay embedding, it is different from the Hankel [4] approach and the time-delay augmented DMD [5]. Unlike the Hankel approach that performs SVD, we design the approach to match the time-delay embedding with a rectangle Koopman matrix directly, which is more efficient for our quantum optimization and quantum machine learning tasks. In addition, we find that the Hankel approach is not stable for quantum optimization applications, while our SW-DMD is more robust. While the time-delay augmented DMD uses time-delay embedding as a data augmentation, it produces a square Koopman operator and may have the difficulty to uniquely determine the prediction in the case of noisy parameter dynamics in optimization tasks on quantum computers. Our approach can uniquely determine the prediction and we also find that it has more stable performance than the time-delay augmented DMD.
> * Our neural network DMDs are designed to be generalizations of the standard DMD and the SW-DMD, so that they agree with DMD and SW-DMD when the neural network becomes an identity function. MLP-DMD and CNN-DMD both generalize from DMD, and MLP-SW-DMD generalizes from SW-DMD. While MLP-DMD only uses the current step parameters information to predict the next step parameters, CNN-DMD uses parameters information in all previous steps to predict the next step parameters. Though our approach is also inspired by the neural network Koopman learning [6,7] (such as Koopman autoencoder), we do not use the autoencoder structure and we find that it also works well for our applications.

---

> ### Author Response · Authors · 2022-11-14
> **Our Work Provides Distinct Contributions for Quantum Optimization and Quantum Machine Learning, instead of a Simple Extension of the Previous Work**
>
> > In my opinion, the strongest point of the paper is the application domain of quantum machine learning (QML) and the incorporation of Koopman-based methodologies in it. Having said that, it should be noted that the technical problem solved in this paper is essentially the ability to approximate complex dynamics with Koopman methods, and the relation to QML is only by application.
> > Unfortunately, there are several weak points. First, the same observation and application was already considered in Dogra et al. (and other papers, some of which cited in this paper). In this context, the main difference between this paper and the paper of Dogra et al. is in the evaluation on QML tasks vs. vision tasks. While this difference may be significant in certain applications and algorithms, I do not believe this is the case here, as they essentially optimize a standard neural network, similar to those considered in Dogra et al.
> > In summary, while the application domain of this paper is interesting and refreshing, it is essentially a different application of an existing paper. Moreover, the proposed methods are similar or equivalent to existing work in the field.
>
> As mentioned in the theoretical and the algorithmic contributions mentioned in the general response above, we would like to further point out that our work is not a simple extension from the previous work.
> * Gradient computation has different complexity in quantum optimization and quantum machine learning compared to the classical case, since it scales with the number of parameters while the classical backpropagation does not. This complexity separation provides more advantages and guarantees for our approach to accelerate quantum optimization and quantum machine learning tasks.
> * Our work applies to not only quantum machine learning, but also quantum optimization. Quantum optimization is a generalization to classical optimization (such as MaxCut [1] and Maximally Independent Set [2]) and it is significant to scientific discoveries. The previous work only studies classical machine learning without consideration of classical optimization tasks.
> * We connect the Koopman Operator theory to the quantum natural gradient in quantum optimization, which does not exist in classical work. It builds a well-motivated theoretical foundation for our work.
> * While our method relies on the ability to approximate complex dynamics with Koopman methods, our work is different from the standard dynamics prediction and the previous one-shot prediction (with no follow-up optimization after the DMD prediction) for classical neural network optimization [3]. The previous classical work only uses standard DMD with no follow-up optimization after the DMD prediction using a large amount of data. However, in quantum optimization and quantum machine learning, the gradient optimization is more challenging and one needs to handle the learning with much fewer training data. Hence, we develop SW-DMD and neural DMD with an iterative optimization protocol tailored for quantum optimization and quantum machine learning. Our methods can be efficiently used with few training data, accelerate the gradient method, and be robust to long-time prediction error and noise. The algorithmic development is new and important to the field.

---

> ### Author Response · Authors · 2022-11-17
> **Looking forward to your feedback**
>
> Dear Reviewer TgmW,
>
> We would be grateful if you can confirm if our response has addressed your concerns and let us know if there is any other question or suggestion. In the following, we summarize the key points of our response:
> * We explain our distinct contributions beyond the previous literature. Theoretically our approach addresses the gradient complexity challenges on quantum computers and connects the Koopman operator theory with quantum natural gradient, both of which do not exist in classical machine learning. Algorithmically we develop new robust iterative DMD algorithms tailored for efficient acceleration of quantum optimization and quantum machine learning.
> * We perform a new series of systematic experiments on various systems of large sizes for both quantum optimization and quantum machine learning, through simulations and real quantum computers.The new results strongly demonstrate the success of our methods.
> * We improve the presentation and discussion of figures, tables and results, which clarifies the confusions in the previous manuscript and makes the paper more comprehensive.
>
> We look forward to hearing your feedback!

---

> ### Author Response · Authors · 2022-12-05
> **Has our response addressed your concerns?**
>
> Dear Reviewer,
>
> Thank you so much for your constructive review which allowed us to improve our paper! In our response, we alleviated potential misunderstandings about the theoretical and algorithmic novelty of our contributions, performed in-depth new experiments, and improved the presentation of the paper. We believe we were able to respond in depth to all of your concerns and kindly ask you to consider increasing your score.
>
> Best wishes,
> The authors

---

### Official Review · Reviewer_XcoP · 2022-10-25

**Confidence:** 3
**Clarity, Quality, Novelty And Reproducibility:** Please refer to the weaknesses section.
**Correctness:** 3
**Technical Novelty And Significance:** 3
**Empirical Novelty And Significance:** 3
**Recommendation:** 6

**Strength And Weaknesses:**

Strength:

The paper discusses an essential problem in the literature.

Also, the author provides an excellent solution to tackle the problem, but the analysis and intuition part is very poorly represented in this paper.

The paper is easy to follow and understand. The authors introduce several quantum machine learning strategies, advantages, and weaknesses. Based on this, the motivation of this paper is reasonable.


Weaknesses:

The author has discussed accelerating quantum mechanics and quantum optimization algorithms compared to classical operators. Did the author quantify or qualify this using some experiment?

How did the Koopman operator help to accelerate the gradient calculation in quantum computers compared to classical computers? Can the author provide a brief explanation for this?

In section5.2, the author has discussed the sequence of the quantum gate. Is this similar to layers? Can the author discuss the objective of each block in Figure 4a (like the objective of block "ansatz" and its input and output)? Similarly need to discuss other blocks. Can the author describe all the blocks and their functionality in the figure-1 for MNIST dataset example (Like, for an image sample)? The author should describe the complete algorithm and discuss which part is classic machine learning and which is quantum machine learning.

Along with MNIST, the author also discuss Figure 1 with the time series task. Did the author report any comparative results on time series?

The author should report the performance and analysis of results on the MNIST dataset.

The author can add a small preliminary section that describes the steps for the quantum ML algorithm (Training and Inference)

The experiment section is not designed correctly. The author should compare the results with SOTA and do an ablation analysis on the Koopman operation. The analysis section of the paper needs to improve with more analysis results.

Take home: Koopman operator  for stability and robustness and
MNIST on QML


**Summary Of The Paper:**

The author has discussed the issue of backpropagation and obtaining the gradient of a quantum computer. The gradient computation during backpropagation in a quantum computer is not similar to a classical computer because the complexity increase with the number of parameters and measurements. The author discusses the Koopman operator theory for predicting nonlinear dynamics with the natural gradient method in quantum optimization to address this issue. The author proposed a data-driven approach using the Koopman operator to accelerate quantum optimization and Quantum machine learning. The author proposed Sliding window dynamic mode decomposition (DMD) and Neural DMD for efficiently updating parameters in quantum computers. The author shows that the method can predict gradient dynamics and accelerate the quantum variational eigensolver in quantum optimization and quantum ML.



**Summary Of The Review:**

The overall writing quality is ok, and the proposed method is simple and beneficial. However, the experiment comparison lacks justification, and the technical contribution is plain.

---

> ### Author Response · Authors · 2022-11-14
> **References**
>
> [1] Akshunna S Dogra and William Redman. Optimizing neural networks via koopman operator theory. Advances in Neural Information Processing Systems, 33:2087–2097, 2020.
>
> [2] Caro, Matthias C., Elies Gil-Fuster, Johannes Jakob Meyer, Jens Eisert, and Ryan Sweke. "Encoding-dependent generalization bounds for parametrized quantum circuits." Quantum 5 (2021): 582.
>
> [3] Ren, Wenhui, Weikang Li, Shibo Xu, Ke Wang, Wenjie Jiang, Feitong Jin, Xuhao Zhu et al. "Experimental quantum adversarial learning with programmable superconducting qubits." arXiv preprint arXiv:2204.01738 (2022).

---

> ### Author Response · Authors · 2022-11-14
> **New Experiments on Quantum Machine Learning with Improved Descriptions and Discussions**
>
> > The author should report the performance and analysis of results on the MNIST dataset.
>
> Yes, we have now included discussion and analysis on the MNIST results, as well as included a table in the appendix to demonstrate the acceleration effect.
>
> > The author can add a small preliminary section that describes the steps for the quantum ML algorithm (Training and Inference)
>
> Yes, we have added some preliminary paragraphs for the quantum ML algorithms and details of our setups in the Appendix L.
>
> > The experiment section is not designed correctly. The author should compare the results with SOTA and do an ablation analysis on the Koopman operation. The analysis section of the paper needs to improve with more analysis results.
>
> Thanks for the suggestion. As mentioned in the general response, we have made the several important updates and would like to point out that
> * We have designed and implemented a series of systematic new experiments with various models of larger system sizes.
> * We have included figures and tables for two ablation studies on quantum Ising model and quantum Heisenberg model.
> * Since quantum machine learning is still in its early exploratory stage, there is no systematic benchmark of methods and SOTA. However, we have included a new quantum machine learning experiment on MNIST with a 10-qubit interleaved quantum circuit. It can be considered as a state-of-the-art quantum machine learning method, since it has been shown to have generalization advantage theoretically [2] and realized recently in a superconducting circuits quantum experiment [3].
> * We have included more analysis for both quantum optimization and quantum machine learning results.
>
> > The overall writing quality is ok, and the proposed method is simple and beneficial. However, the experiment comparison lacks justification, and the technical contribution is plain.
>
> With the general response above, we would like to point out that
> * We have designed and implemented a series of systematic new experiments with various models of larger system sizes. All the new experiments have strongly supported that our approach is efficient for accelerating quantum optimization and quantum machine learning.
> * We have developed SW-DMD and neural DMD with an iterative optimization protocol with the goal of accelerating quantum optimization and quantum machine learning. This is different from both the standard dynamics prediction and the previous one-shot prediction (with no follow-up optimization after the DMD prediction) for classical neural network optimization [1]. Our methods can be efficiently used with few training data, accelerate the gradient method, and be robust to long-time prediction error and noise. The algorithmic development is new and important to the field.

---

> ### Author Response · Authors · 2022-11-14
> **Improved Presentation on the Results and the Analysis**
>
> > Also, the author provides an excellent solution to tackle the problem, but the analysis and intuition part is very poorly represented in this paper.
>
> Thanks for the comments. As mentioned in the general response, we have now included more discussions on our motivations, theoretical and algorithmic development, result analysis and presentation.
>
> > The author has discussed accelerating quantum mechanics and quantum optimization algorithms compared to classical operators. Did the author quantify or qualify this using some experiment?
>
> As mentioned in the general response above, we have implemented a variety of new experiments over different models of larger system sizes. In particular, we include a) quantum Ising model with different couplings and quantum Heisenberg model, b) increase natural gradient simulations from 5 qubits to 10 qubits, c) increase Adam simulations from 10 qubits to 12 qubits, d) include 5-qubit noisy adam simulation, e) increase real quantum computer experiments from 3 qubits to 5 qubits, f) increase quantum machine learning simulations from 2 qubits to 10 qubits. We also include tables to precisely demonstrate the acceleration effects of our approach for ablation studies and quantum machine learning. All the new experiments have strongly supported that our approach is efficient for accelerating quantum optimization and quantum machine learning.
>
> > How did the Koopman operator help to accelerate the gradient calculation in quantum computers compared to classical computers? Can the author provide a brief explanation for this?
>
> As mentioned in the theoretical contribution in the general response above, the gradient computation has a complexity separation between quantum computers and classical computers. On quantum computers, to compute gradients one needs to do it for each parameter with sampling, which scales with both the number of parameters and the number of measurements per parameter. On classical computers, gradient computation can be done with backward propagation, which has the same complexity as the forward evaluation which does not scale with the number of parameters. By using the Koopman operator, we can predict the gradient update on a classical computer and verify its performance on a quantum computer directly. The parameter verification on a quantum computer only scales with the number of measurements and does not depend on the number of parameters, so that it can accelerate the gradient calculation.
>
> > In section5.2, the author has discussed the sequence of the quantum gate. Is this similar to layers? Can the author discuss the objective of each block in Figure 4a (like the objective of block "ansatz" and its input and output)? Similarly need to discuss other blocks. Can the author describe all the blocks and their functionality in the figure-1 for MNIST dataset example (Like, for an image sample)? The author should describe the complete algorithm and discuss which part is classic machine learning and which is quantum machine learning.
>
> Thanks for the suggestions. The sequence of the quantum gate can be viewed as layers. We have also included detailed discussion on various perspectives of the quantum machine learning setup in the Appendix M. In the hybrid classical-quantum machine learning task, the latent-space image data x are fed into the quantum circuit via the FeatureMap gate, as the FeatureMap as a layer is a function of x. The ansatz layer plays the same role as the ansatz layer in the quantum optimization case, where theta is the parameter in the layer that we need to optimize. Finally the quantum measurement turns the quantum output data into classical outputs.
>
> In addition, we now perform a new quantum machine learning experiment with 10 qubits using a state-of-the-art quantum circuit, details of which are also mentioned in the Appendix L. The new QML experiment is purely quantum without a classical neural network, and the interlayered encoding block is a function of both input data x and parameters theta and cannot be separated into the two consecutive blocks as FeatureMap followed by ansatz.
>
> > Along with MNIST, the author also discuss Figure 1 with the time series task. Did the author report any comparative results on time series?
>
> We would like to clarify that the “time series” in Fig. 1 indicates that we view the gradient dynamics as a time series so that we can apply Koopman operator theory for prediction. While we are not performing time series tasks with quantum machine learning in this work, our approach is generic and can be applied to accelerate time series applications as well. This will be an interesting direction for future study.

---

> ### Author Response · Authors · 2022-11-17
> **Looking forward to your feedback**
>
> Dear Reviewer XcoP,
>
> We would be grateful if you can confirm if our response has addressed your concerns and let us know if there is any other question or suggestion. In the following, we summarize the key points of our response:
> * We explain our distinct contributions beyond the previous literature. Theoretically our approach addresses the gradient complexity challenges on quantum computers and connects the Koopman operator theory with quantum natural gradient, both of which do not exist in classical machine learning. Algorithmically we develop new robust iterative DMD algorithms tailored for efficient acceleration of quantum optimization and quantum machine learning.
> * We perform a new series of systematic experiments on various systems of large sizes for both quantum optimization and quantum machine learning, through simulations and real quantum computers.The new results strongly demonstrate the success of our methods.
> * We improve the presentation and discussion of figures, tables and results, which clarifies the confusions in the previous manuscript and makes the paper more comprehensive.
>
> We look forward to hearing your feedback!

---

> ### Author Response · Authors · 2022-12-05
> **Has our response addressed your concerns?**
>
> Dear Reviewer,
>
> Thank you so much for your constructive review which allowed us to improve our paper! In our response, we highlighted the theoretical and algorithmic novelty of our contributions, performed in-depth new experiments, and improved the presentation of the paper. We believe we were able to respond in depth to all of your concerns and kindly ask you to consider increasing your score.
>
> Best wishes,
> The authors

---

### Official Review · Reviewer_5VYL · 2022-10-26

**Confidence:** 4
**Correctness:** 3
**Technical Novelty And Significance:** 2
**Empirical Novelty And Significance:** 2
**Recommendation:** 3

**Clarity, Quality, Novelty And Reproducibility:**

The paper is clear and well written. The ide is a simple extension of classical work to the quantum case. The results are reproducible

**Strength And Weaknesses:**

+ the Koopam operator optimization method may turn out to be interesting for learning
- there is hardly any strong evidence in this work of the performance of the methods
- the experiments are trivial (3 qubits) and no conclusions can be drawn from them
- no theoretical reasons why this methods should outperform other gradient-free optimization methods are given

**Summary Of The Paper:**

The authors present two algorithms for quantum optimization and machine learning based on the Koopman operator. They test their algorithm with simulations and a small hardware experiment.

**Summary Of The Review:**

The paper has an interesting idea of using Koopman operator theory as a way to avoid gradient calculations in training quantum neural networks. More work and evidence is needed for the impact of this methods

---

> ### Author Response · Authors · 2022-11-14
> **References**
>
> [1] H Li, Z Xu, G Taylor, C Studer, T Goldstein. Visualizing the Loss Landscape of Neural Nets. Neural Information Processing Systems (NeurIPS).
>
> [2] Nesterov, Y., Spokoiny, V. Random Gradient-Free Minimization of Convex Functions. Found Comput Math 17, 527–566 (2017). https://doi.org/10.1007/s10208-015-9296-2
>
> [3] James Martens. New insights and perspectives on the natural gradient method. Journal of Machine Learning Research 21 (2020) 1-76.
>
> [4] Ryan Sweke, Frederik Wilde, Johannes Meyer, Maria Schuld, Paul K. Faehrmann, Barthélémy Meynard-Piganeau, Jens Eisert. Stochastic gradient descent for hybrid quantum-classical optimization. Quantum 4, 314 (2020).
>
> [5] Harrigan, M.P., Sung, K.J., Neeley, M. et al. Quantum approximate optimization of non-planar graph problems on a planar superconducting processor. Nat. Phys. 17, 332–336 (2021). https://doi.org/10.1038/s41567-020-01105-y.
>
> [6] S. Ebadi, A. Keesling, M. Cain, T. T. Wang, H. Levine, D. Bluvstein, G. Semeghini, A. Omran, J.-G. Liu, R. Samajdar, X.-Z. Luo, B. Nash, X. Gao, B. Barak, E. Farhi, S. Sachdev, N. Gemelke, L. Zhou, S. Choi, H. Pichler, S.-T. Wang, M. Greiner, V. Vuleti ́c , and M. D. Lukin. Quantum optimization of maximum independent set using rydberg atom arrays. Science, 376(6598): 1209–1215, jun 2022. doi: 10.1126/science.abo6587. URL https://doi.org/10.1126% 2Fscience.abo6587.

---

> ### Author Response · Authors · 2022-11-14
> **Our Work Provides Distinct Contributions for Quantum Optimization and Quantum Machine Learning, instead of a Simple Extension of the Classical Work**
>
> > The idea is a simple extension of classical work to the quantum case.
>
> With the theoretical and the algorithmic contributions mentioned in the general response above, we would like to further point out that our work is not a simple extension from the previous work.
> * Gradient computation has different complexity in quantum optimization and quantum machine learning compared to the classical case, since it scales with the number of parameters while the classical backpropagation does not. This complexity separation provides more advantages and guarantees for our approach to accelerate quantum optimization and quantum machine learning tasks.
> * Our work applies to not only quantum machine learning, but also quantum optimization. Quantum optimization is a generalization to classical optimization (such as MaxCut [5] and Maximally Independent Set [6]) and it is significant to scientific discoveries. The previous work only studies classical machine learning without consideration of classical optimization tasks.
> * We connect the Koopman Operator theory to the quantum natural gradient in quantum optimization, which does not exist in classical work. It builds a well-motivated theoretical foundation for our work.
> * The previous classical work only uses standard DMD with no follow-up optimization after the DMD prediction using a large amount of data. However, we notice that this could not be applied to quantum optimization and quantum machine learning with much fewer training data. Hence, we develop SW-DMD and neural DMD with an iterative optimization protocol tailored for quantum optimization and quantum machine learning. Our methods can be efficiently used with few training data, accelerate the gradient method, and be robust to long-time prediction error and noise. The algorithmic development is new and important to the field.

---

> ### Author Response · Authors · 2022-11-14
> **Theoretical Support for Koopman Operator Learning being better than the Gradient-free Optimization**
>
> > no theoretical reasons why this methods should outperform other gradient-free optimization methods are given
>
> Thanks for the comment, and we would like to point out that our approach has theoretical advantages over gradient-free optimization. The reasoning comes in two steps:
> * For high dimensional optimization, the gradient method is known to have better performance than gradient-free methods.  When gradients of the objective function are known, we have access to information about the local geometry [1], and thus gradient-based algorithms alone are known to converge much faster to optima compared to gradient-free algorithms, both theoretically and empirically [2]. Furthermore, the natural gradient method takes further advantage from the Fisher information matrix for accessing the curvature information, and it is shown to have a strong convergence property [3]. In particular for quantum optimization, the gradient method has been shown to converge [4]. Meanwhile, the gradient-free method provides noisy signals for the loss decay and it becomes more challenging for high dimensional optimization.
> * Our approach performs as well as the gradient method even in the worst case and can perform better than the gradient method in practice. This is because our approach predicts the gradient method with an iterative optimization protocol to select optimal prediction. At the zeroth order, our approach follows the gradient dynamics which has advantages over gradient-free methods as discussed above. In particular, in quantum optimization, the Koopman operator is naturally connected to the quantum natural gradient, which is a powerful gradient method for optimization. With the iterative protocol, we can make long time predictions and choose the optimal parameters for the next round of gradient optimization. Due to the complexity separation between gradient computation and prediction verification on quantum computers, the prediction verification cost is negligible. This guarantees that our approach will not be affected by the long-time prediction error and noise, and it should be at least as good as the previous gradient optimization result. Since our approach accelerates over the gradient method, it will be better than the gradient-free optimization.

---

> ### Author Response · Authors · 2022-11-14
> **Theoretical and Experimental Evidence for the Success of the Koopman Operator Learning**
>
> > there is hardly any strong evidence in this work of the performance of the methods
>
> With our improved presentation and new experiments, we would like to emphasize that
> * Theoretically, we connect the Koopman operator theory with the quantum natural gradient in quantum optimization, which provides a well-grounded foundation of our method.
> * Algorithmically, we develop SW-DMD and neural network DMD with an iterative optimization protocol for quantum optimization, which goes beyond the standard DMD. Due to the complexity separation between gradient computation and prediction verification on quantum computers, our approach offers more guarantees and advantages for acceleration and is robust against long-time prediction error and noise.
> * Experimentally, we have implemented a series of new experiments on various systems of larger scale systems for both quantum optimization and quantum machine learning, which is mentioned in the general response above. All the experiments have demonstrated significant acceleration of our methods on the tasks.
>
> We are also sorry that the presentation of figures may be misleading in the previous manuscript, especially the plot of gradient and prediction in Fig.3 which shows the long-time blow up prediction. As mentioned in the general response above, we would like to point out that the prediction is much less costly compared to the gradient and they should be plotted together. The long-time error of prediction is  also not an issue under our iterative optimization algorithm since the optimal prediction can be used for the initial point in the next-round gradient optimization. Hence, we now only plot the gradient update with iterations and use a dashed line to indicate the acceleration effect of the predictions. We also include new figures and tables for ablation studies for precise demonstration on the acceleration effect of different methods.
>
> > the experiments are trivial (3 qubits) and no conclusions can be drawn from them
>
> The experiment on the real IBM Lima quantum computer was on 3 qubits in our previous manuscript, but we also have numerical simulations of larger systems which also give promising results. In the current era, quantum computing resources are still limited and expensive, and the 3-qubit real experiment, despite being a small system, can still provide valuable guidance with realistic experimental issues for further practice. We now have 5-qubit real quantum computer experiments in the updated manuscript.
>
> As is mentioned in the general response above, we have implemented a variety of new experiments over different models of larger system sizes. In particular, we include a) quantum Ising model with different couplings and quantum Heisenberg model, b) increase natural gradient simulations from 5 qubits to 10 qubits, c) increase Adam simulations from 10 qubits to 12 qubits, d) include 5-qubit noisy Adam simulation, e) increase real quantum computer experiments from 3 qubits to 5 qubits, f) increase quantum machine learning simulations from 2 qubits to 10 qubits. We also include tables to precisely demonstrate the acceleration effects of our approach for ablation studies and quantum machine learning. All the new experiments have strongly supported that our approach is efficient for accelerating quantum optimization and quantum machine learning.

---

> ### Author Response · Authors · 2022-11-17
> **Looking forward to your feedback**
>
> Dear Reviewer 5VYL,
>
> We would be grateful if you can confirm if our response has addressed your concerns and let us know if there is any other question or suggestion. In the following, we summarize the key points of our response:
> * We explain our distinct contributions beyond the previous literature. Theoretically our approach addresses the gradient complexity challenges on quantum computers and connects the Koopman operator theory with quantum natural gradient, both of which do not exist in classical machine learning. Algorithmically we develop new robust iterative DMD algorithms tailored for efficient acceleration of quantum optimization and quantum machine learning.
> * We perform a new series of systematic experiments on various systems of large sizes for both quantum optimization and quantum machine learning, through simulations and real quantum computers.The new results strongly demonstrate the success of our methods.
> * We improve the presentation and discussion of figures, tables and results, which clarifies the confusions in the previous manuscript and makes the paper more comprehensive.
>
>  We look forward to hearing your feedback!

---

> ### Author Response · Authors · 2022-12-05
> **Has our response addressed your concerns?**
>
> Dear Reviewer,
>
> Thank you so much for your constructive review which allowed us to improve our paper! In our response, we alleviated potential misunderstandings about the theoretical and algorithmic novelty of our contributions, performed in-depth new experiments, and improved the presentation of the paper. We believe we were able to respond in depth to all of your concerns and kindly ask you to consider increasing your score.
>
> Best wishes,
> The authors

---

### Author Response · Authors · 2022-11-14
**General Response: Theoretical and Algorithmic Contributions to Accelerating Quantum Optimization and Quantum Machine Learning**

We would like to thank the reviewers for all of their comments and the suggestions. There are experiments and descriptions in the original manuscript that could be further improved. Based on the reviewers’ feedback, we have discussed and highlighted the theoretical and algorithmic contributions of our work, performed new systematic experiments of larger scale in both simulations and the real IBM Lima quantum computer, and included better figures, tables and description for presenting the results. We believe we have responded in depth to all of the concerns of the reviewers and we kindly ask the reviewers to consider increasing their scores.

First of all, we would like to clarify that our work is not a simple extension of Dogra et al.’s work [1]. Our work has both new theoretical and algorithmic contributions. Theoretically, we would like to point out
* The gradient computation between classical computation and quantum computation has complexity separation. While backward propagation does not scale with the number of parameters, the computation of gradient in quantum computation scales linearly with the number of parameters. While this difference makes quantum optimization and quantum machine learning much more challenging than the classical counterparts, it implies that our acceleration algorithms have more complexity guarantees and advantages compared to the classical case, which is particularly important to the field.
* While our work has applications to quantum machine learning, another important component of the work is related to quantum optimization, which is not discussed in the previous literature.  Even though quantum optimization starts with a quantum Hamiltonian, it is a generalization of classical optimization. Any classical optimization problem can be encoded into a quantum Hamiltonian such that quantum computers can be used to solve the problem, and the lowest energy solution is the optimal solution for the original problem. Recently, encouraging efforts on quantum computation have been demonstrated on important classical optimization, such as MaxCut [2] and Maximally Independent Set [3]. Scientifically, quantum optimization is crucial for new chemistry and materials discoveries, since quantum effects are intrinsic in the setups. For accelerating the above applications, our algorithms will be beneficial and provide powerful tools.  Hence, our work has broad implications not only to quantum machine learning, but also to both classical and quantum optimizations.
* Though the Koopman operator theory can be applied to classical neural network optimization in an existing manner without an explicit constructive proof at a theoretical level [1], the theoretical connection is much more natural to quantum optimization, as we demonstrate with an explicit construction in our work. One theoretical contribution from our work is to explicitly relate the quantum natural gradient to the Koopman operator theory, which does not exist in classical machine learning. The connection provides a nice theoretical foundation for our work and it has been demonstrated successfully in the natural gradient experiments.

Algorithmically, compared to the standard DMD for optimizing classical neural network, we want to emphasize that
* Our work has developed sliding window DMD and neural network DMD with the goal for accelerating quantum optimization and quantum machine learning, which yields a novel continuation of the algorithmic advancements from the previous work [1]. Our experiments have shown that the sliding window DMD and neural network DMD have much better performances than the standard DMD in general.
* While there are developments on time-delay DMD and neural network DMD for classical dynamics prediction, our algorithms have certain technical distinctions for efficiency and stability which are now discussed in the manuscript, and more importantly, our focus is on optimization instead of dynamics prediction. Since gradient optimization is more challenging in quantum computation compared to classical optimization, we notice that the one-shot prediction approach (with no follow-up optimization after the DMD prediction) with training data, consisting of many parameters, in the previous work [1] does not work well for accelerating quantum optimization and machine learning. Hence, we develop the iterative DMD optimization protocol shown in Fig. 1 in the manuscript. Due to the complexity separation between gradient computation and prediction verification on quantum computers, we can verify our prediction of parameters for many step updates efficiently and choose the optimal one. This algorithm can effectively accelerate the optimization with a few training data records and is robust against long-time prediction error and noise, which is crucial for realistic quantum optimization and machine learning.

---

> ### Author Response · Authors · 2022-11-14
> **General Response: New Series of Systematic Experiments on Various Systems of Larger Sizes**
>
> To further solidify our approach, we have designed and implemented a variety of new experiments as follows. All the experiments have consistenetly demonstrated that our methods are successful and promising.
> * A new model: we include natural gradient and Adam simulations for the quantum Heisenberg model at Jz=0.5, which is a more complicated model. The additional experiments broaden the scope of our work from the previous work with the quantum Ising model only.
> * Larger scale simulations: our work has increased the simulations for natural gradient method from 5 qubits to 10 qubits and Adam method from 10 qubits to 12 qubits. The larger system sizes can give more convincing results. Increasing the system size in simulations to even more qubits is challenging because of the inherent exponential scaling in a classical computer to simulate a quantum computer.
> * Larger scale real IBM quantum computer experiments: even though the access to real quantum computers is expensive, we have extended our experiments from a 3-qubit IBM quantum computer with SW-DMD to a 5-qubit IBM quantum computer with SW-DMD and MLP-SW-DMD.
> * Larger scale quantum machine learning simulations: besides the previous quantum machine learning application using 2-qubit quantum circuit, we have implemented experiments on a 10-qubit interleaved quantum circuit with 270 parameters. The new quantum circuit is more realistic, which has been shown to have generalization advantage [4] and is used in recent experiments in superconducting circuits [5].
> * Noisy simulations with Adam: to test the robustness of our approach, we have carried out Adam simulations with a noisy quantum circuit model on 5 qubits with various measurement shots. It is shown that our method can achieve acceleration under different noisy conditions which are of interest in realistic experiments in the near-term quantum computers.
> * Simulations on sliding window effects: to further understand the performance of the sliding window DMD, we also include experiments for different sliding window sizes and figure out the optimal choice of sliding window size.
> * In our framework of alternating VQE+DMD runs, we now also include the last iteration of VQE when determining t_opt, so that even in extreme cases where the DMD predictions are inaccurate, the performance of our scheme will be at least as good as the traditional VQE.

---

> ### Author Response · Authors · 2022-11-14
> **General Response: Improved Presentation of Figures, Tables and Results**
>
> To better illustrate our algorithms and results, we have also made the following updates on our presentation in the manuscript.
> * Previously we have plotted both the gradient update and the prediction in the same figure (such as Fig. 3). We find that this might be misleading since the gradient update and the prediction has a complexity separation on quantum computers, where the latter one does not scale with the number of parameters and has much smaller cost. Moreover, the long time prediction blow-up in the previous figure is actually not an issue in practice since our algorithm always chooses the optimal parameters as the initial point for next-round gradient update. Therefore, we now use the word “iteration” to indicate gradient interaction step, which is the dominant cost in quantum optimization and machine learning. We only plot the gradient iterations with the solid line and use the dashed line to indicate the acceleration effect of the DMD predictions. We further illustrate with concrete examples and figures on how our iteration algorithm works and how the optimal parameters from the prediction are chosen in the appendix.
> * We have included new tables for the ablation tasks and the quantum machine learning application to precisely demonstrate the acceleration effects of our approach.
> * We have included more discussions on the theoretical and algorithmic contributions of our work, the interpretation of our results, the noise analysis of the experiment on the real quantum computer IBM Lima, the background of quantum machine learning and quantum computation in the main text and the appendix. We hope that the new manuscript is more comprehensive and informative.

---

> ### Author Response · Authors · 2022-11-14
> **General Response: References**
>
> [1] Akshunna S Dogra and William Redman. Optimizing neural networks via koopman operator theory. Advances in Neural Information Processing Systems, 33:2087–2097, 2020.
>
> [2] Harrigan, M.P., Sung, K.J., Neeley, M. et al. Quantum approximate optimization of non-planar graph problems on a planar superconducting processor. Nat. Phys. 17, 332–336 (2021). https://doi.org/10.1038/s41567-020-01105-y.
>
> [3] S. Ebadi, A. Keesling, M. Cain, T. T. Wang, H. Levine, D. Bluvstein, G. Semeghini, A. Omran, J.-G. Liu, R. Samajdar, X.-Z. Luo, B. Nash, X. Gao, B. Barak, E. Farhi, S. Sachdev, N. Gemelke, L. Zhou, S. Choi, H. Pichler, S.-T. Wang, M. Greiner, V. Vuleti ́c , and M. D. Lukin. Quantum optimization of maximum independent set using rydberg atom arrays. Science, 376(6598): 1209–1215, jun 2022. doi: 10.1126/science.abo6587. URL https://doi.org/10.1126% 2Fscience.abo6587.
>
> [4] Caro, Matthias C., Elies Gil-Fuster, Johannes Jakob Meyer, Jens Eisert, and Ryan Sweke. "Encoding-dependent generalization bounds for parametrized quantum circuits." Quantum 5 (2021): 582.
>
> [5] Ren, Wenhui, Weikang Li, Shibo Xu, Ke Wang, Wenjie Jiang, Feitong Jin, Xuhao Zhu et al. "Experimental quantum adversarial learning with programmable superconducting qubits." arXiv preprint arXiv:2204.01738 (2022).

---

### Author Response · Authors · 2022-12-07
**We hope to engage in constructive discussion before the review period ends**

Dear reviewers,

We hope to engage in constructive discussion before the discussion period ends since we have addressed all your comments in our response. It will be appreciated to have your feedback and thanks for your support.

Best,
The authors

---

### Decision · Program_Chairs · 2023-01-20

**Decision:**

Reject

**Justification For Why Not Higher Score:**

Two reviewers vote strongly for rejection with high confidence.

**Justification For Why Not Lower Score:**

N/A

**Metareview: Summary, Strengths And Weaknesses:**


Summary:

The paper discusses backpropagation and gradient computation in a quantum computer. The gradient computation in a quantum computer has a complexity that increases with the number of parameters and measurements. The paper discusses the Koopman operator theory for predicting nonlinear dynamics with the natural gradient method in quantum optimization to address this issue. The paper proposes a data-driven approach using the Koopman operator to accelerate quantum optimization and Quantum machine learning. The paper shows that the method can predict gradient dynamics and accelerate the quantum variational eigensolver in quantum optimization and quantum ML.

Strengths:

- the Koopam operator optimization method may turn out to be interesting for learning
- the paper discusses an essential problem in the literature.
- the author provides an excellent solution to tackle the problem, but the analysis and intuition part is very poorly represented in this paper.
- The paper is easy to follow and understand. The authors introduce several quantum machine learning strategies, advantages, and weaknesses. Based on this, the motivation of this paper is reasonable.
- the application domain of quantum machine learning (QML) and the incorporation of Koopman-based methodologies in it

Weaknesses:

- there is hardly any strong evidence in this work of the performance of the methods
- the experiments are trivial (3 qubits) and no conclusions can be drawn from them
- no theoretical reasons why this methods should outperform other gradient-free optimization methods are given
- the technical problem solved in this paper is essentially the ability to approximate complex dynamics with Koopman methods, and the relation to QML is only by application
- the same observation and application was already considered in Dogra et al.
- the proposed approaches to approximate the Koopman operator generally ignore many similar if not identical ideas in the literature

Recommendation:

Some reviewers vote slightly toward acceptance and others vote strongly for rejection. The reviewers with the most confident decisions vote for rejection. In a comment submitted during the time to write the meta-review, the authors point out concerns regarding the reviews from reviewers 5VYL and TgmW, indicating that they have misunderstood the theoretical and algorithmic contributions of the paper. If such were the case, that would indicate that the paper would still need more work to clearly present such theoretical and algorithmic contributions in a better way. Based on all this, I have decided to recommend rejection and encourage the authors to use the feedback provided to improve the paper and resubmit to another venue.

---

> ### Author Response · Authors · 2023-01-31
> **Response to the paper decision**
>
> It is a pity to hear about the final decision, which is solely based on the reviews from the first round, without any further discussion and taking any account of our response that addresses all the reviewers’ concerns.
>
> You mention the following weaknesses:
>
> > there is hardly any strong evidence in this work of the performance of the methods
>
> This is a copy of 5VYL’s comment. We disagree. See our reply to 5VYL. We have made extensive experiments in our work and theoretical justification.
>
> > the experiments are trivial (3 qubits) and no conclusions can be drawn from them
>
> This is a copy of 5VYL’s comment. The claim is false. See our reply to 5VYL. We have experiments on up to 12 qubits.
>
> > no theoretical reasons why this methods should outperform other gradient-free optimization methods are given
>
> This is a copy of 5BYL’s comment. The claim is false. See our reply to 5VYL.
>
> > the technical problem solved in this paper is essentially the ability to approximate complex dynamics with Koopman methods, and the relation to QML is only by application
>
> This is a copy of TgmW’s comment. The claim is an attempt to trivialize our work, and we disagree with it. See our reply to the reviewer.
>
> > the same observation and application was already considered in Dogra et al.
>
> This is a copy of TgmW’s comment. The claim is false. See our reply to TgmW.
>
> > the proposed approaches to approximate the Koopman operator generally ignore many similar if not identical ideas in the literature
>
> This is a copy of TgmW’s comment. We have responded to it thoroughly. See our reply to TgmW.
>
> We have repeatedly raised our concern about the misunderstandings of Reviewers 5VYL and TgmW to the Area Chairs even before the time to write the meta-review. We have not received any reply during the discussion period, only an acknowledgment from TgmW that they have “read the authors’ responses.”
>
> Thank you for suggesting presenting our contributions in a better way. We will follow your advice. We believe we have made the paper clearer during the discussion period. It’s a pity that our effort was not acknowledged.
>
> From the authors.